

Geoscientific
Model Development

# On numerical broadening of particle-size spectra: a condensational growth study using PyMPDATA 1.0

**Michael A. Olesik**[1], **Jakub Banaśkiewicz**[2], **Piotr Bartman**[2], **Manuel Baumgartner**[3,4], **Simon Unterstrasser**[5], **and Sylwester Arabas**[6,2]

[1]Faculty of Physics, Astronomy and Applied Computer Science, Jagiellonian University, Kraków, Poland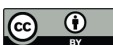
[2]Faculty of Mathematics and Computer Science, Jagiellonian University, Kraków, Poland
[3]Zentrum für Datenverarbeitung, Johannes Gutenberg University Mainz, Mainz, Germany
[4]Institute for Atmospheric Physics, Johannes Gutenberg University Mainz, Mainz, Germany
[5]Institute of Atmospheric Physics, German Aerospace Center (DLR), Oberpfaffenhofen, Germany
[6]Department of Atmospheric Sciences, University of Illinois at Urbana-Champaign, Urbana, IL, USA

**Correspondence:** Michael A. Olesik (michael.olesik@doctoral.uj.edu.pl) and Sylwester Arabas (sylwester.arabas@uj.edu.pl)

**Abstract.** TS2 This work discusses the numerical aspects of representing the condensational growth of particles in models of aerosol systems such as atmospheric clouds. It focuses on the Eulerian modelling approach, in which fixed-bin discretisation is used for the probability density function describing the particle-size spectrum. Numerical diffusion is inherent to the employment of the fixed-bin discretisation for solving the arising transport problem (advection equation describing size spectrum evolution). The focus of this work is on a technique for reducing the numerical diffusion in solutions based on the upwind scheme: the multidimensional positive definite advection transport algorithm (MPDATA). Several MPDATA variants are explored including infinite-gauge, non-oscillatory, third-order terms and recursive antidiffusive correction (double-pass donor cell, DPDC) options. Methodologies for handling coordinate transformations associated with both particle-size spectrum coordinate choice and with numerical grid layout choice are expounded. Analysis of the performance of the scheme for different discretisation parameters and different settings of the algorithm is performed using (i) an analytically solvable box-model test case and (ii) the single-column kinematic driver ("KiD") test case in which the size-spectral advection due to condensation is solved simultaneously with the advection in the vertical spatial coordinate, and in which the supersaturation evolution is coupled with the droplet growth through water mass budget. The box-model problem covers size-spectral dynamics only; no spatial dimension is considered. The single-column test case involves a numerical solution of a two-dimensional advection problem (spectral and spatial dimensions). The discussion presented in the paper covers size-spectral, spatial and temporal convergence as well as computational cost, conservativeness and quantification of the numerical broadening of the particle-size spectrum. The box-model simulations demonstrate that, compared with upwind solutions, even a 10-fold decrease in the spurious numerical spectral broadening can be obtained by an apt choice of the MPDATA variant (maintaining the same spatial and temporal resolution), yet at an increased computational cost. Analyses using the single-column test case reveal that the width of the droplet size spectrum is affected by numerical diffusion pertinent to both spatial and spectral advection. Application of even a single corrective iteration of MPDATA robustly decreases the relative dispersion of the droplet spectrum, roughly by a factor of 2 at the levels of maximal liquid water content. Presented simulations are carried out using PyMPDATA – a new open-source Python implementation of MPDATA based on the Numba just-in-time compilation infrastructure.

**Published by Copernicus Publications on behalf of the European Geosciences Union.**

## 1 Introduction

### 1.1 Motivation and outline

The focus of this paper is on the problem of particle-size evolution for a population of droplets undergoing condensational growth. Representing the particle-size spectrum using a number density function, the problem can be stated as a population-balance equation expressing conservation of the number of particles. Herein, the numerical solution of the problem using the MPDATA family of finite difference schemes originating in Smolarkiewicz (1983, 1984) is discussed. MPDATA stands for multidimensional positive definite advection transport algorithm and is a higher-order iterative extension of the forward-in-time upwind scheme. Iterative application of upwind scheme, first using the physical velocity and subsequently with so-called antidiffusive velocities, results in MPDATA being characterised by small amounts of implicit diffusion but retaining the salient features of the upwind scheme: conservativeness, small phase error and sign preservation.

MPDATA features a variety of options allowing an algorithm variant that is appropriate to the problem at hand to be picked. This work highlights the importance of the MPDATA algorithm variant choice for the resultant spectral broadening of the particle-size spectrum. The term spectral broadening refers to the increase in width of the droplet spectrum during the lifetime of a cloud. The broadening may be associated with both physical mechanisms (including turbulent mixing, particle composition diversity, radiative heat transfer effects; see for example Feingold and Chuang, 2002) as well as with spurious artefacts stemming from the employed numerical solution technique, the latter being the focus of this work.

Cloud simulations with a detailed treatment of droplet microphysics face a twofold challenge in resolving the droplet spectrum width. First, it is challenging to model and numerically represent the subtleties of condensational growth which link the physico-chemical properties of single particles with ambient thermodynamics through latent heat release and multi-particle competition for available vapour (e.g. Arabas and Shima, 2017; Yang et al., 2018), even more so when considering the interplay between particle growth and supersaturation fluctuations (e.g. Jeffery et al., 2007; Abade et al., 2018). Second, the discretisation strategies employed in representing the particle-size spectrum and its evolution are characterised by inherent limitations which constrain the fidelity of spectral width predictions (e.g. Arabas and Pawlowska, 2011; Morrison et al., 2018). Finally, corroboration of spectral width estimates from both theory and modelling against experimental data faces the problems of instrumental broadening inherent to the measurement techniques (e.g. Devenish et al., 2012, Sect. 3.2) and the problem of sampling volume choice (e.g. Kostinski and Jameson, 2000).

The width of the spectrum plays a key role in determining both the droplet collision probabilities (Grabowski and Wang, 2013) and the characteristics relevant for radiative transfer (Chandrakar et al., 2018). These in turn are reflected in parameterisations of cloud processes in large-scale models. Taking climate-timescale simulation as an example, the representation of clouds remains the largest source of uncertainty (Schneider et al., 2017). Parameterisations used in climate models, for example of such cloud microphysical processes as cloud condensation nuclei (CCN) activation, are developed based on smaller-scale simulations resolving particle-size spectrum evolution. Consequently, it is of importance to quantify the extent to which the droplet-size spectrum width is a consequence of (a) the physics of particle growth embodied in the governing equations and (b) the discretisation and the associated numerical diffusion.

The following introductory subsections start with a chronologically presented literature review of applications of MPDATA to the problem of condensational growth of particles. Section 2 focuses on a simple analytically solvable box-model test case with no spatial dimension considered and serves as a tutorial on MPDATA variants. It is presented to gather the information that is scattered across works focusing on more complex computational fluid dynamics applications of MPDATA. Presented simulations pertaining to the evolution of cloud droplet size spectrum in a cumulus cloud depict how enabling the discussed algorithm variants affects simulated droplet spectrum width. An analysis of the computational cost of different algorithm variants is carried out and corroborated with previously published data. While comprehensive from the point of view of the considered problem of diffusional growth (and hence limited to one-dimensional homogeneous advection of positive-sign fields), the presented material merely hints at the versatility of the algorithm. For a proper review of the MPDATA family of algorithms highlighting the multi-dimensional aspects and its multifaceted applications, we refer to Smolarkiewicz and Margolin (1998), Smolarkiewicz (2006), and Kühnlein and Smolarkiewicz (2017).

Section 3 covers the application of MPDATA for coupled size-spectral and spatial advection in a single-column kinematic set-up from Shipway and Hill (2012). First, the methodology to handle the spectral-spatial liquid water advection problem taking into account the coupling with the vapour field is detailed. Second, the results obtained using different MPDATA variants are discussed focusing on the measures of spectral broadening.

Section 4 concludes the work with a summary of findings. Appendix A contains a convergence analysis based on results of multiple simulations using the box-model test case run with different temporal and spatial (size-spectral) resolutions and compared with the analytical solution.

All presented simulations are performed with PyMPDATA (Bartman et al., 2022a) – an open-source Python implementation of MPDATA built on top of the Numba just-in-time

compilation framework. In terms of numerics, PyMPDATA follows libmpdata++ (Jaruga et al., 2015).

## 1.2 Background

There exist two contrasting approaches for modelling the evolution of droplet-size spectrum (see Grabowski, 2020, for a review): the Eulerian (fixed-bin) and the Lagrangian (moving-bin, moving-sectional or particle-based). Overall, while the Lagrangian methods are the focus of active research and development (Morrison et al., 2020), the Eulerian schemes have been predominantly used in large-scale modelling (Khain et al., 2015).

Following Liu et al. (1997) and Morrison et al. (2018), the earliest documented study employing the Eulerian numerics for condensational growth of a population of particles is that of Kovetz and Olund (1969). Several earlier works, starting with the seminal study of Howell (1949), utilised the Lagrangian approach. The numerical scheme proposed in Kovetz and Olund (1969, Eq. 10) resembles an upwind algorithm, being explicit in time and orienting the finite-difference stencil differently for condensation and evaporation.

An early discussion of numerical broadening of the cloud droplet spectrum can be found in Brown (1980) where the numerical scheme from Kovetz and Olund (1969) was improved in several ways, including the sampling of the drop growth rate at the bin boundaries (as is done herein). Brown (1980) also covers quantification of the error of the method by comparisons to analytical solutions.

In Tsang and Brock (1982), an Eulerian–Lagrangian scheme is considered that combines a method-of-characteristics solution with spline interpolation onto a fixed grid. Based on comparison with the Eulerian-Lagrangian scheme, the authors conclude that the upwind differencing is not suitable for aerosol growth calculations due to its unacceptable numerical diffusion. It is worth noting that the study includes considerations of the Kelvin effect of surface tension on the drop growth (not considered herein; see discussion of Eq. 5 below).

The first mention of an application of the MPDATA scheme for the problem of condensational growth can be found in Smolarkiewicz (1984). The problem is given as an example where the divergent-flow option of the algorithm may be applicable (see Sect. 2.8 below).

In Tsang and Korgaonkar (1987), which is focused on evaporation, MPDATA is used as a predictor step followed by a corrective step using a Galerkin finite-element solver. In two subsequent studies from the same group (Tsang and Rao, 1988, 1990), MPDATA is compared to other algorithms in terms of conservativeness and computational cost. In Tsang and Rao (1988), the basic three-iteration MPDATA was used. Intriguingly, it is noted there that "If the antidiffusion velocities are increased by some factor between 1.04 and 1.08, use of [corrective iteration] only once can reduce 50 % of the computing time [...] without much sacrifice of accuracy". In conclusion, the authors praised MPDATA for providing narrow size spectra. Tsang and Rao (1988) pointed out that MPDATA performs worse than the upwind scheme in terms of the prediction of the mean radius.

The "Aerosol Science: Theory and Practice" book of Williams and Loyalka (1991) contains a section (5.19) on MPDATA (termed "Smolarkiewicz method") within a chapter focusing on the methods of solving the dynamic equation describing aerosol spectrum evolution. The basic variant of MPDATA (Smolarkiewicz, 1983) is presented with an outline of its derivation.

In Kostoglou and Karabelas (1995) and Dhaniyala and Wexler (1996), the authors mention that MPDATA has the potential to reduce the numerical diffusion as compared to the upwind scheme in the context of particle-size evolution calculations. However, the high computational cost of the method is given as a reason to discard the scheme from further analysis.

In Morrison et al. (2018), a comparison of different numerical schemes for the condensational growth problem is performed. Fixed-bin and moving-bin approaches are compared. MPDATA (referred to as MPDG therein) with the non-oscillatory option enabled is reported to produce most significant numerical diffusion and spectral broadening among employed fixed-bin methods. Intriguingly, as can be seen in Fig. 7 therein, the broad spectrum appears as early as after 20 time steps, at an altitude of 20 m (out of 520 m of the simulated parcel displacement). Thus, it is questionable if the broadening attributed to the diffusivity of MPDATA is not an artefact of the top-hat initial condition.

In Wei et al. (2020), MPDATA is employed for integrating droplet spectrum evolution for comparison with a Lagrangian scheme. The work concludes that the spurious broadening of the spectrum cannot be alleviated even with a spectral discretisation involving 2000 size bins.

The discussion presented in Morrison et al. (2018) prompted further analyses presented in Hernandéz Pardo et al. (2020) and Lee et al. (2021). These studies highlighted that, in principle, the problem is a four-dimensional transport problem (three spatial dimensions and the spectral dimension) and that the interplay of spectral and spatial advection further nuances the issue of spectral broadening.

It is worth noting that none of the works mentioned above discussed coordinate transformations to non-linear grid layouts with MPDATA (a discussion of handling non-uniform mesh with the upwind scheme can be found in Li et al., 2017, their Appendix A). Wei et al. (2020) and Morrison et al. (2018) are the only works mentioning any options other than the basic flavour of the scheme, yet only the non-oscillatory option was considered. Herein, the applicability for solving the condensational growth problem of multiple variants of MPDATA and their combinations is expounded.

## 1.3 Governing equations

To describe the conservation of particle number $N$ under the evolution of the particle-size spectrum $n_p(p) = \frac{dN}{dp}$ ($n$ denoting number density as a function of particle-size parameter $p$ such as radius or volume), one may take the one-dimensional continuity equation (i.e. Liouville equation expressing the conservation of probability; for discussion see Hulburt and Katz, 1964), in a generalised coordinate system:

$$\partial_t(G n_p) + \partial_x(u G n_p) = 0, \tag{1}$$

where $G \equiv G(x)$ represents the coordinate transformation from $p$ to $x$ with $x$ being an equidistant mesh coordinate used in the numerical solution, $n_p \equiv n_p(p(x))$ being number density function and $u \equiv u(x)$ denoting the pace of particle growth in the chosen coordinate $x$. Equation (1) in this context is also referred to as a population balance equation (see for example Ramkrishna, 2000, chap. 2.7).

The coordinate transformation term $G$ may play a 2-fold role in this context. First, one has the choice of the particle-size parameter used as the coordinate (i.e. the argument $p$ of the density function $n(p)$). For the chosen coordinates $p \in [r, s \sim r^2, v \sim r^3]$, the appropriate distributions will be $n_r(r)$, $n_s(s)$ and $n_v(v)$, where $s = 4\pi r^2$ and $v = 4/3\,\pi r^3$ denote particle surface and volume, respectively. The size spectrum $n_p(p)$ in a given coordinate is related with $n_r(r)$ via the following relation of measures: $n_p(p)dp = n_r(r)dr$ such that the total number $N = \int n_r dr$ is conserved. Second, one has the choice of the grid layout $p(r(x))$, that is, how the parameters $r$, $s$ or $v$ are discretised to form the equidistant grid in $x$. This can be used, for instance, to define a mass-doubling grid layout ($x = \ln_2(r^3)$) as used in Morrison et al. (2018) and herein.

Combination of the two transformations yields the following definition of $G$:

$$G \equiv dp(r)/dx(r) = \frac{dp}{dx}, \tag{2}$$

which defines the transformation from the coordinate $p$ of the density function to the numerical mesh coordinate $x$. For further discussion of the coordinate transformation approaches in the embraced framework (including multi-dimensional setting), see Smolarkiewicz and Clark (1986) and Smolarkiewicz and Margolin (1993).

## 2 Spectral advection with upwind and MPDATA solutions (box-model test case)

### 2.1 Upwind discretisation

The numerical solution of Eq. (1) is obtained on a grid defined by $x = i \cdot \Delta x$ and at discrete time steps defined by $t = n \cdot \Delta t$. Henceforth, $\psi_i^n$ and $G_i$ denote the discretised number density $n_p$ and the discretised coordinate transformation term, respectively. The dimensionless advective field is denoted by $GC = \frac{dp}{dx} u \Delta t / \Delta x$, where $C$ stands for the Courant number, i.e. the velocity in terms of temporal and spatial grid increments. Note that the values of the Courant number itself are not used, only the product GC (of the coordinate transformation term $G$ and the Courant number $C$) is used. A staggered grid is employed and indicated with fractional indices for vector fields, e.g. $GC_{i+1/2} \equiv (GC)|_{i+1/2}$ in the case of the discretisation of GC. A finite-difference form of the differential operators is introduced embracing the so-called upwind approach (dating back at least to Courant et al., 1952, Eq. 16 therein):

$$\psi_i^{n+1} = \psi_i^n - \frac{1}{G_i}\left(F(\psi_i^n, \psi_{i+1}^n, GC_{i+1/2}) - F(\psi_{i-1}^n, \psi_i^n, GC_{i-1/2})\right) \tag{3}$$

with TS3

$$F(\psi_L, \psi_R, GC_{mid}) = \max(GC_{mid}, 0) \cdot \psi_L + \min(GC_{mid}, 0) \cdot \psi_R, \tag{4}$$

where the introduced flux function $F$ defines the flux of $\psi$ across a grid-cell boundary. Hereinafter a shorthand notation $F_{i+\frac{1}{2}}(\psi) \equiv F(\psi_i, \psi_{i+1}, GC_{i+\frac{1}{2}})$ is used.

### 2.2 Box-model test case and upwind solution

The test case is based on Fig. 3 from East (1957) – one of the early papers on the topic of cloud droplet spectral broadening. The case considers the growth of a population of cloud droplets through condensation in the equilibrium supersaturation limit, where

$$u \approx \frac{dx}{dr}\dot{r} = \frac{dx}{dr}\frac{\xi}{r}, \tag{5}$$

with $\xi = \xi_0(S - 1)$ being an approximately constant factor proportional to the supersaturation $(S - 1)$ where the saturation $S$ denotes the relative humidity of ambient air. The parameter $\xi_0$ is set to $100\,\mu m^2\,s^{-1}$ to match the results from East (1957). The approximation (Eq. 5) neglects the dependence of particle growth rate on the surface tension (Kelvin term). Taking it into consideration requires replacing $(S - 1)$ with $(S - e^{A/r})$, where $A$ depends on temperature only; for discussion see for example Tsang and Brock (1982).

For the initial number density function, an idealised fair-weather cumulus droplet size spectrum is modelled with a lognormal distribution:

$$n_r^{(0)}(r) = n_0 \exp\left(-\kappa(\log_{10}(r/r_0))^2\right)/r, \tag{6}$$

with $r_0 = 7\,\mu m$ and $\kappa = 22$ (East and Marshall, 1954), and $n_0 = 465\,cm^{-3}$ to match the liquid water content of $1\,g\,kg^{-1}$ as indicated in East (1957).

For the boundary conditions (implemented using halo grid cells), linear extrapolation is applied to $G$, while both $\psi$ and GC are set to zero within the halo.

Analytical solution to Eq. (1) is readily obtainable for $\dot{r} = \xi/r$ and for any initial size spectrum. Noting that introducing $x = r^2$ coordinates, the transport Eq. (1) becomes a constant-coefficient advection equation; the problem reduces to translation in $x$ by $2\xi t$. Cast in the $r$ coordinate, the solution can be expressed as follows (Kovetz, 1969):

$$\psi^{\text{analytical}} = n_{\text{r}}(r, t > 0) \equiv \frac{r}{\tilde{r}} n_{\text{r}}^{(0)}(\tilde{r}), \tag{7}$$

where $\tilde{r} = \tilde{r}(r, t) = \sqrt{r^2 - 2\xi t}$.

The upper panels in Figs. 1 and 2 depict the droplet size spectrum evolution through condensational growth from an initial liquid water mixing ratio of $M_0 = 1\,\text{g}\,\text{kg}^{-1}$ under supersaturation of $S - 1 = 0.075\,\%$.

Two grid layout $(x)$ and size parameter $(p)$ choices are depicted. Figure 1 presents a simulation carried out with a $p = r^2$ coordinate and discretised on a mass-doubling grid $(x = \ln_2(r^3))$. Figure 2 presents simulation results obtained with $x = r$ and $p = r$. In all cases, the time step is set to $\Delta t = \frac{1}{3}$ s. The domain span is 1–26 µm. The grid is composed of 75 grid cells. Such settings correspond to $GC \approx 0.26$ for $p = r^2$, and a variable Courant number approximately in the range of 0.03 to 0.07 for $p = r$.

The times for which results are depicted in the plots are selected by finding $t$ for which the integrated liquid water mixing ratio of the analytical solution is equal to 1, 4 and $10\,\text{g}\,\text{kg}^{-1}$ (assuming air density of $1\,\text{kg}\,\text{m}^{-3}$). In both Figs. 1 and 2, the upper panels show the number density and the bottom panels show the normalised mass density. The bottom panels thus depict the same quantities as Fig. 3 in East (1957). A similar comparison of upwind and analytical solutions is also presented in Fig. 1a in Li et al. (2017).

The normalised mass density of bin $i$ is evaluated as $4/3\pi \rho_L m_i^{(l=3)}/M$, where $\rho_L = 1000\,\text{kg}\,\text{m}^{-1}$, by calculating the third statistical moment of the number distribution $n_{\text{r}}(p)$ with the formula:

$$m_i^{(l)} = \int_{r_1}^{r_2} n_{\text{r}} r^l \mathrm{d}r = $$

$$= \psi_i \cdot \begin{cases} (l+1)^{-1} r^{l+1} \big|_{r_1}^{r_2} & \text{for } p = r \\ 2(l+2)^{-1} (r^2)^{\frac{l+2}{2}} \big|_{r_1^2}^{r_2^2} & \text{for } p = r^2, \end{cases} \tag{8}$$

where $r_1$, $r_2$ are the boundaries of $i$th bin, and $\psi_i$ is the value of $n_{\text{p}}$ associated with the bin (i.e. $n_{\text{p}}$ is assumed to be binwise constant; note that the physical unit associated with $n_{\text{p}}$ depends on the choice of $p$). The normalisation factor $M$ is the water mixing ratio (e.g. $M = M_0 = 1\,\text{g}\,\text{kg}^{-1}$ for $t = 0$).

Looking at the mass density plots in Figs. 1 and 2, it is evident that casting the results in the form of mass density shifts positions of the extrema in comparison with the analytical solution. This is one of the consequences of solving the

number conservation equation (for discussion see Sect. 2.12). As can be seen in both the number- and mass-density plots in Figs. 1 and 2, solutions obtained with the upwind scheme are characterised by a significant drop in the peak value and visible broadening of the spectrum with respect to the analytical solution – both manifesting the numerical diffusion. The broadening and the drop in the peak value are less pronounced in Fig. 2 where employment of a linear grid causes an increase in the resolution in the large-particle region of the spectrum as compared to mass-doubling grid case of Fig. 1.

## 2.3 Truncation error analysis of the upwind scheme

One of the methods used to quantify the numerical diffusion of a numerical scheme is the modified equation analysis of Hirt (1968) (see Margolin and Shashkov, 2006, for discussion in the context of MPDATA). In a nutshell, the analysis involves (i) Taylor expansion of each term of the numerical scheme; (ii) elimination of higher-than-first-order time derivatives using the time-differentiated original advection equation; and (iii) derivation of a partial differential equation, referred to as the modified equation, that a given scheme actually approximates in lieu of the advection equation.

To depict an application of the modified equation analysis in the present context (upwind scheme), a simplified setting where $G = 1$ and $C$ is constant is outlined herein. In the analysis, the Taylor expansion of $\psi$ up to the second order is taken at $\psi_i^{n+1}$, $\psi_{i+1}^n$ and $\psi_{i-1}^n$ and substituted into the numerical upwind scheme, in which the flux function (Eq. 4) is expressed using moduli (e.g. Crowley, 1968, Eq. 12):

$$\psi_i^{n+1} = \psi_i^n - \left( \frac{C + |C|}{2} (\psi_i^n - \psi_{i-1}^n) + \frac{C - |C|}{2} (\psi_{i+1}^n - \psi_i^n) \right), \tag{9}$$

which results in

$$\partial_t \psi + \partial_t^2 \psi \frac{\Delta t}{2} = -\frac{u + |u|}{2} \left( \partial_x \psi - \partial_x^2 \psi \frac{\Delta x}{2} \right) - \frac{u - |u|}{2} \left( \partial_x \psi + \partial_x^2 \psi \frac{\Delta x}{2} \right), \tag{10}$$

which can be further transformed by employing a time derivative of both sides of the original advection equation $\partial_t \psi = -u \partial_x \psi \longrightarrow \partial_t^2 \psi = -u \partial_x \partial_t \psi = u^2 \partial_x^2 \psi$ to substitute the second-order time derivative with a spatial derivative (Cauchy–Kowalevski procedure; see Toro, 1999) leading to the sought modified equation (Roberts and Weiss, 1966, Eq. 2.9):

$$\partial_t \psi + u \partial_x \psi + \underbrace{\left( u^2 \frac{\Delta t}{2} - |u| \frac{\Delta x}{2} \right)}_{K} \partial_x^2 \psi + \ldots = 0. \tag{11}$$

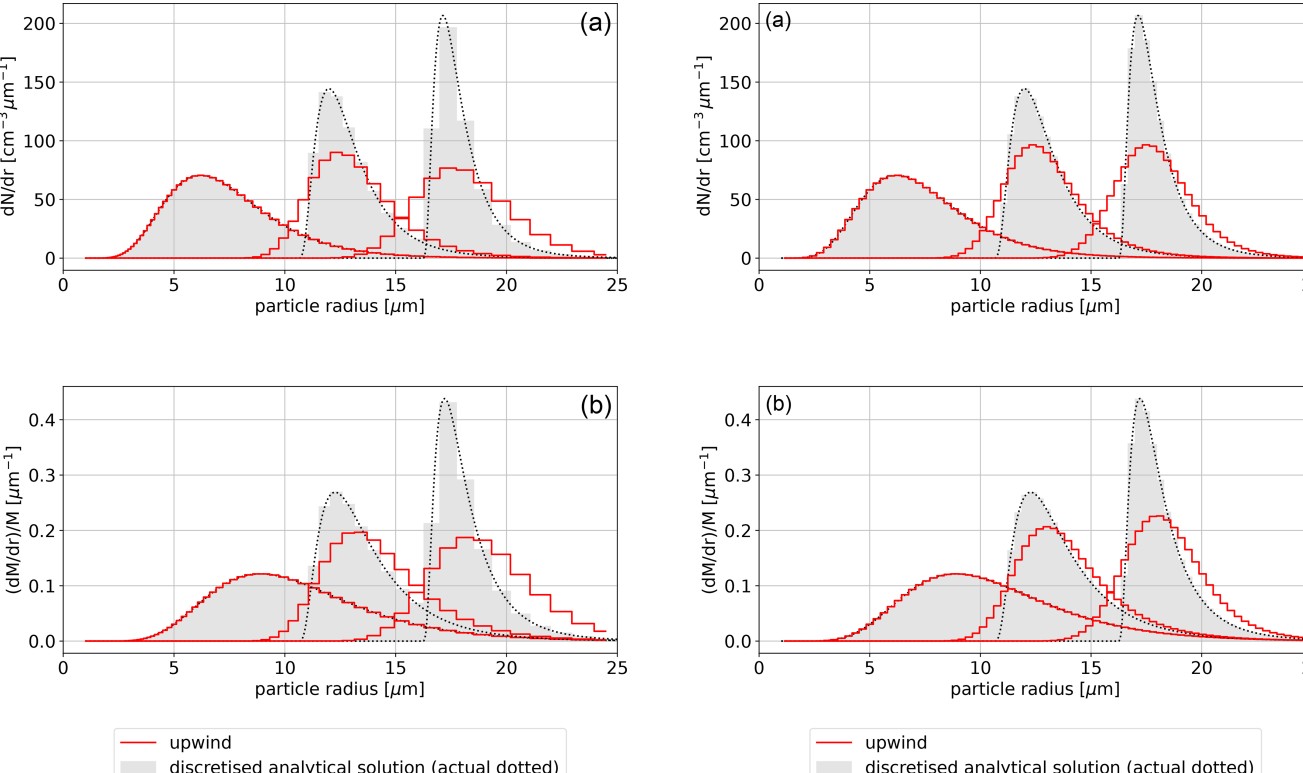

**Figure 1.** Evolution of the particle number density (**a**) and normalised mass density (**b**) with red histograms corresponding to the numerical solution using upwind scheme, black dots depicting analytical solution, and grey-filled histograms representing discretised analytical solution; compare with Fig. 3 in East (1957). Numerical solution was obtained with $p = r^2$ on a mass-doubling grid, i.e. with $x = \ln_2(r^3)$.

The above analysis depicts that the employment of the numerical scheme (Eq. 3) results in a solution of a modified equation (Eq. 11), approximating the original problem up to first order. The leading second-order error contribution has the form of a diffusive term with a coefficient $K$ (note that the above outline of the modified equation analysis assumes a constant velocity field). The diffusive form of the leading error term explains the smoothing of the spectrum evident in Figs. 1 and 2 and is consistent with the notion of numerical diffusion.

### 2.4 Antidiffusive velocity and iterative corrections

The problem of numerical diffusion can be addressed by introducing the so-called "antidiffusive velocity" (Smolarkiewicz, 1983). To this end, the Fickian flux can be cast in the form of an advective flux – an approach dubbed pseudo-velocity technique in the context of advection–diffusion simulations (Lange, 1973, 1978) or hyperbolic formulation of diffusion (Cristiani, 2015, discussion of Eq. 4 therein) – and is discussed in detail in Smolarkiewicz and Clark (1986,

**Figure 2.** As in Fig. 1 for $p = r$ and $x = r$.

Sect. 3.2):

$$\partial_x (K \partial_x \psi) = \partial_x \left( K \frac{\partial_x \psi}{\psi} \psi \right). \tag{12}$$

In Smolarkiewicz (1983, 1984), it was proposed to apply the identity (Eq. 12) to Eq. (11) to suppress the spurious diffusion. The procedure is iterative. The first iteration is the basic upwind pass. Subsequent corrective iterations reverse the effect of numerical diffusion by performing upwind passes with the so-called antidiffusive flux based on Eq. (12) but with $K$ taken with a negative sign and approximated using the upwind stencil (for discussion of the discretisation, see Smolarkiewicz and Margolin, 2001).

Accordingly, the basic antidiffusive field $GC^{(k)}$ is defined as follows (with $\epsilon > 0$ being an arbitrarily small constant used to prevent from divisions by zero):

$$GC^{(k)}_{i+\frac{1}{2}} = A_{i+\frac{1}{2}} \left( \left| GC^{(k-1)}_{i+\frac{1}{2}} \right| - \left( GC^{(k-1)}_{i+\frac{1}{2}} \right)^2 \right), \tag{13}$$

where $k$ is the iteration number, $GC^{(1)} \equiv GC$ and

$$A_{i+\frac{1}{2}} = \frac{\psi^*_{i+1} - \psi^*_i}{\psi^*_{i+1} + \psi^*_i + \epsilon}, \tag{14}$$

where $\psi^*$ denotes $\psi^n$ in the first iteration, or the values resulting from the application of the upwind scheme with

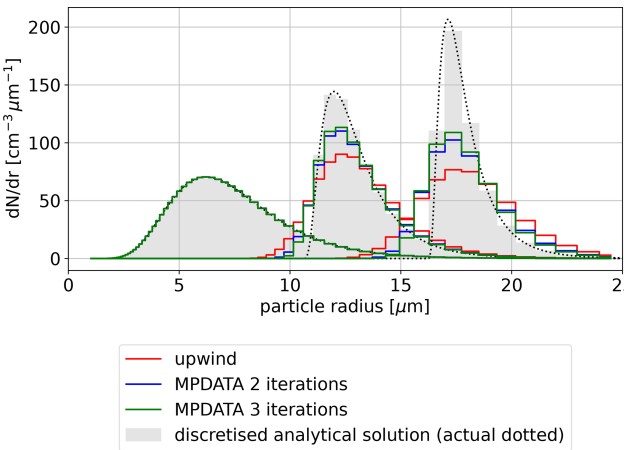

**Figure 3.** Comparison of analytical, upwind and MPDATA solutions (see plot key for algorithm variant specification) using the setup from Fig. 1; see Sect. 2.4 for discussion.

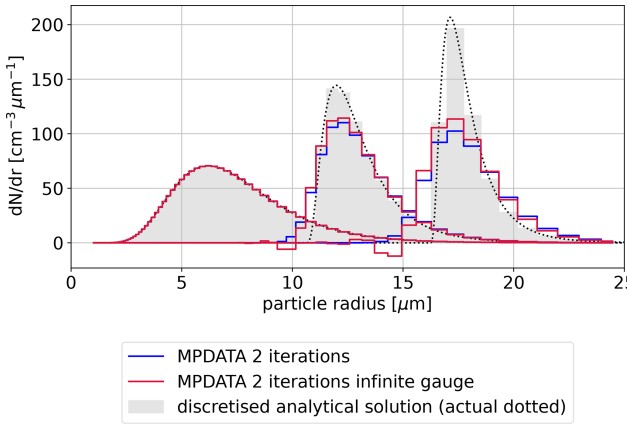

**Figure 4.** Comparison of analytical, upwind and MPDATA solutions (see plot key for algorithm variant specification) using the setup from Fig. 1; see Sect. 2.5 for discussion.

the antidiffusive flux in subsequent iterations. The MPDATA scheme inherits the key properties of the upwind scheme in terms of positive-definiteness, conservativeness and stability while reducing the effect of numerical diffusion. In all presented formulæ below, it is assumed that $\psi$ is positive (as in the case of particle number density). The references provided include formulation of the algorithm for variable-sign fields (e.g. momentum advection).

In Fig. 3, the analytical results obtained with upwind solutions and presented in Fig. 1 are supplemented with results obtained using the MPDATA scheme with two and three iterations. Employment of MPDATA corrects (with respect to analytical solution) both the peak amplitude and the spectrum width, as well as the position of the maximum. It is visible that the effect of the third iteration is less pronounced than that of the second one. Overall, while the MPDATA solutions are superior to the upwind solutions, the drop in amplitude and broadening of the resultant spectrum still visibly differs from the discretised analytical solution.

## 2.5 Infinite gauge variant

For the possible improvement of the algorithm, one may consider linearising MPDATA about an arbitrarily large constant (i.e. taking $\psi' = \psi + a\chi$ in the limit $a \longrightarrow \infty$ instead of $\psi$, where $\chi$ is a constant scalar background field). Such analysis was considered in Smolarkiewicz and Clark (1986, Eq. 41) and subsequently referred to as the "infinite-gauge" (or "iga") variant of MPDATA (Smolarkiewicz, 2006, their Eq. 34; Margolin and Shashkov, 2006, point (6) on page 1204).

Such gauge transformation changes the corrective iterations of the basic algorithm as follows (replacing Eqs. 14 and

4, which is symbolised with $\rightsquigarrow$):

$$A_{i+\frac{1}{2}} \quad \rightsquigarrow \quad A_{i+\frac{1}{2}}^{(\text{iga})} = \frac{\psi_{i+1}^* - \psi_i^*}{2}, \tag{15}$$

$$F_{i+\frac{1}{2}} \quad \rightsquigarrow \quad F_{i+\frac{1}{2}}^{(\text{iga})} = G C_{i+\frac{1}{2}}^{(k)}. \tag{16}$$

Note that the amplitude of the diffusive flux (Eq. 12) is inversely proportional to the amplitude of the transported field. Consequently, such a gauge choice decreases the amplitude of the truncation error (see Smolarkiewicz and Clark, 1986, p. 408, Jaruga et al., 2015, discussion of Fig. 11). However, the infinite-gauge variant no longer assures positive-definite solutions.

Figure 4 depicts how enabling the infinite gauge variant influences results presented in Fig. 3. In each plotted time step, the maximum amplitude of the infinite-gauge result is closest to the analytical solution – improving over the basic MPDATA. However, in each case, negative values are also observed (non-physical in case of the considered problem). Consequently, for the problem at hand, it is effectively essential to combine it with the monotonicity-preserving non-oscillatory option outlined in the next section.

## 2.6 Non-oscillatory option

In Smolarkiewicz and Grabowski (1990), an extension of the MPDATA algorithm was introduced that makes the solution monotonicity-preserving. In the case of the infinite-gauge variant outlined above, it precludes the appearance of negative values in the discussed solution of droplet-size spectrum evolution. The trade-off is that the order of the algorithm is reduced (see Appendix A).

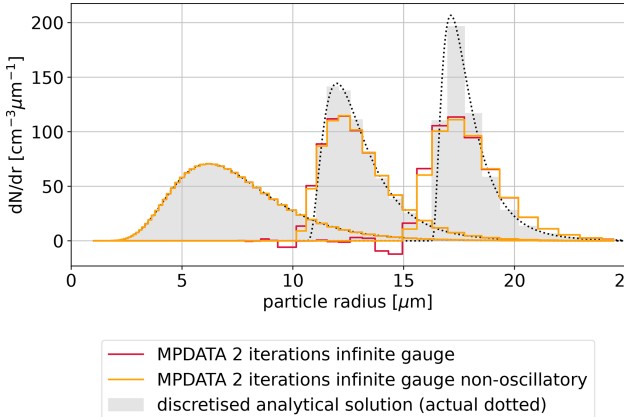

**Figure 5.** Comparison of analytical, upwind and MPDATA solutions (see plot key for algorithm variant specification) using the set-up from Fig. 1; see Sect. 2.6 for discussion.

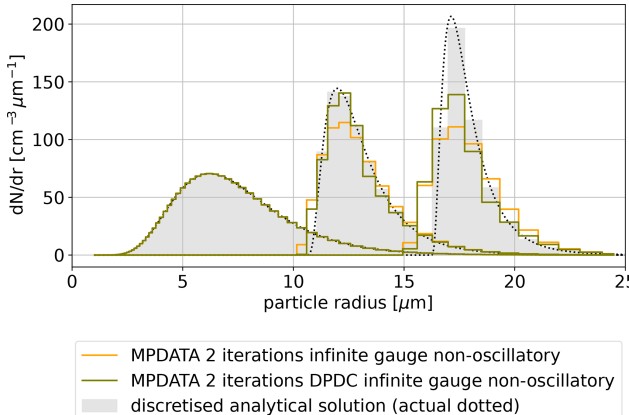

**Figure 6.** Comparison of analytical, upwind and MPDATA solutions (see plot key for algorithm variant specification) using the set-up from Fig. 1; see Sect. 2.7 for discussion.

The non-oscillatory option (later referred to as "non-osc" herein) modifies the algorithm as follows:

$$
GC_{i+\frac{1}{2}}^{(k+1)} \rightsquigarrow GC_{i+\frac{1}{2}}^{(k+1,\text{non-osc})} = GC_{i+\frac{1}{2}}^{(k)} \times
$$
$$
\times \begin{cases} \min(1, \beta_i^{\downarrow}, \beta_{i+1}^{\uparrow}) & GC_{i+\frac{1}{2}}^{(k)} \geq 0 \\ \min(1, \beta_i^{\uparrow}, \beta_{i+1}^{\downarrow}) & GC_{i+\frac{1}{2}}^{(k)} < 0 \end{cases}, \quad (17)
$$

where

$$
\beta_i^{\uparrow} \equiv G_i \times \frac{\max\left(\psi_i^{(\text{max})}, \psi_{i-1}^*, \psi_i^*, \psi_{i+1}^*\right) - \psi_i^*}{\max\left(F(\psi^*)_{i-\frac{1}{2}}, 0\right) - \min\left(F(\psi_i^*)_{i+\frac{1}{2}}, 0\right) + \epsilon}, \quad (18)
$$

and

$$
\beta_i^{\downarrow} \equiv G_i \times \frac{\min\left(\psi_i^{(\text{min})}, \psi_{i-1}^*, \psi_i^*, \psi_{i+1}^*\right) - \psi_i^*}{\max\left(F(\psi_i^*)_{i+\frac{1}{2}}, 0\right) - \min\left(F(\psi_i^*)_{i-\frac{1}{2}}, 0\right) + \epsilon}, \quad (19)
$$

with

$$
\psi_i^{(\text{min})} = \min(\psi_{i-1}^n, \psi_i^n, \psi_{i+1}^n),
$$
$$
\psi_i^{(\text{max})} = \max(\psi_{i-1}^n, \psi_i^n, \psi_{i+1}^n). \quad (20)
$$

Note that in the case of the infinite gauge option being enabled, $F$ function takes the form presented in Eq. (15) (see also Hill, 2011, Sect. 2.5).

Figure 5 juxtaposes infinite-gauge solutions with the non-oscillatory option switched on or off. The effectiveness of the latter variant is apparent as spurious negative values no longer occur.

## 2.7 DPDC

An alternative to the iterative application of the antidiffusive velocities was introduced in Beason and Margolin (1988);

Margolin and Smolarkiewicz (1998) and further discussed in Margolin and Shashkov (2006), where the contributions of multiple corrective iterations of MPDATA were analytically summed. This leads to a new two-pass scheme dubbed DPDC (double-pass donor cell), featuring the following form of the antidiffusive GC field:

$$
GC_{i+\frac{1}{2}}^{(2)} \rightsquigarrow GC_{i+\frac{1}{2}}^{(\text{DPDC})} = \frac{GC^{(2)}}{1 - |A_{i+\frac{1}{2}}|} \left(1 - \frac{GC^{(2)}}{1 - A_{i+\frac{1}{2}}^2}\right), \quad (21)
$$

with $A_{i+\frac{1}{2}}$ defined in Eq. (14). Note that only one corrective iteration is performed with the DPDC variant.

An example simulation combining the double-pass (DPDC), the non-oscillatory and infinite-gauge variants is presented in Fig. 6, which depicts how the solution is improved over that in Fig. 5.

## 2.8 Divergent-flow correction

For divergent flows (hereinafter abbreviated dfl), the modified equation analysis yields an additional correction term in the antidiffusive velocity formula (see Smolarkiewicz, 1984, Eq. 38, for uniform coordinates; Margolin and Smolarkiewicz, 1998, Eq. 30, for non-uniform coordinates; and Waruszewski et al., 2018, Sect. 4, for the infinite-gauge variant):

$$
GC_{i+\frac{1}{2}}^{(k)} \rightsquigarrow GC_{i+\frac{1}{2}}^{(k,\text{dfl})} = GC^{(k)} - \frac{GC_{i+\frac{1}{2}}^{(k)}}{G_{i+1} + G_i} \times
$$
$$
\times \frac{GC_{i+\frac{3}{2}}^{(k)} - GC_{i-1/2}^{(k)}}{2} \times
$$
$$
\times \begin{cases} (\psi_{i+1}^* + \psi_i^*)/2 & (\text{iga}) \\ 1 & (\text{else}). \end{cases} \quad (22)
$$

As pointed out in Sect. 5.1 in Smolarkiewicz (1984), this option has the potential of improving results for the problem of the evolution of the droplet size spectrum (personal communication with William Hall cited therein). This is due to the drop growth velocity defined by Eq. (5) being dependent on the droplet radius and hence divergent. Yet, applying adequate coordinate transformation (i.e. $p = r^2$), the drop growth velocity in the transformed coordinates becomes constant (see Sect. 2.2 above; see for example Hall, 1980, Sect. 3b). Nevertheless, the antidiffusive velocities employed in corrective iterations of MPDATA are in principle divergent; hence the option has the potential to influence results even with $p = r^2$.

In simulations using the presented set-up (also for $p \neq r^2$; not shown), only insignificant changes in the results when the divergent-flow option was used were observed. However, the problem considered herein does not include, for instance, the surface tension influence on the drop growth rate.

## 2.9 Third-order terms

Another possible improvement to the algorithm comes from the inclusion of the third-order terms in the modified equation analysis, which leads to the following form of the antidiffusive velocity (Margolin and Smolarkiewicz, 1998):

$$GC_{i+\frac{1}{2}}^{(k)} \rightsquigarrow GC_{i+\frac{1}{2}}^{(k,\text{tot})} = GC^{(k)} + B_i \cdot GC_{i+\frac{1}{2}}^{(k)} \times$$

$$\times \frac{1}{6}\left(4\frac{|GC_{i+\frac{1}{2}}^{(k)}|}{G_{i+1}+G_i} - 8\left(\frac{GC_{i+\frac{1}{2}}^{(k)}}{G_{i+1}+G_i}\right)^2 - 1\right), \quad (23)$$

$$B_i = 2\cdot(\psi_{i+2}^* - \psi_{i+1}^* - \psi_i^* + \psi_{i-1}^*) \times$$

$$\times \begin{cases} (1+1+1+1)^{-1} & \text{(iga)} \\ (\psi_{i+2}^* + \psi_{i+1}^* + \psi_i^* + \psi_{i-1}^*)^{-1} & \text{(else)}. \end{cases} \quad (24)$$

Figure 7 depicts how enabling the third-order terms improves the solution with respect to the upwind and basic MPDATA solutions.

It is worth noting that discussion of higher-order variants of MPDATA was carried forward in Kuo et al. (1999) and Waruszewski et al. (2018). In the latter case, the focus was placed on accounting for coordinate transformation and variable velocity in the derivation of antidiffusive velocities leading to a fully third-order accurate scheme.

## 2.10 A "best" combination of options

The MPDATA variants presented in the preceding sections can be combined together. In Fig. 8, results obtained with the upwind scheme and with the basic two-pass MPDATA are compared to those obtained with three iterations and the third-order terms, the infinite-gauge and the non-oscillatory options enabled simultaneously. This combination of options

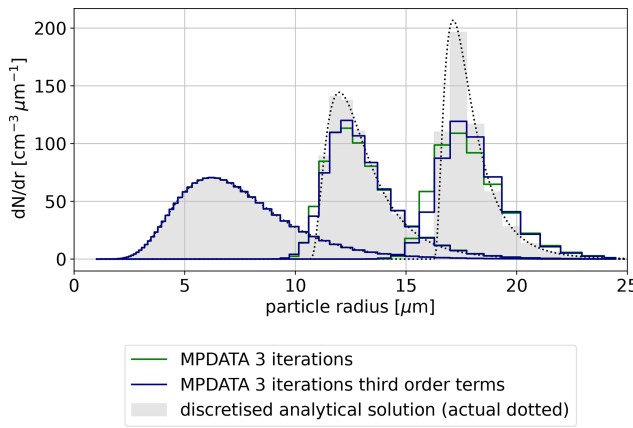

**Figure 7.** Comparison of analytical, upwind and MPDATA solutions (see plot key for algorithm variant specification) using the set-up from Fig. 1; see Sect. 2.9 for discussion.

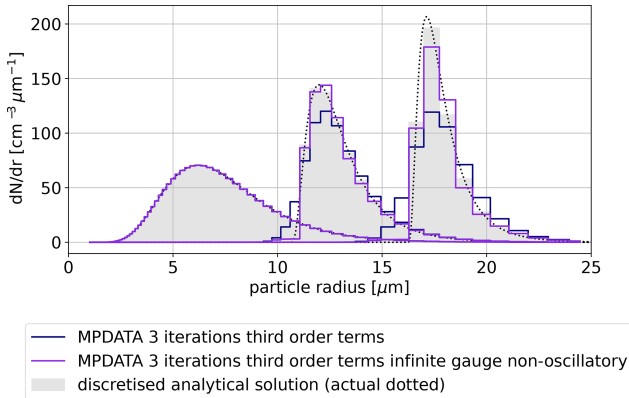

**Figure 8.** Comparison of analytical, upwind and MPDATA solutions (see plot key for algorithm variant specification) using the set-up from Fig. 1; see Sect. 2.10 for discussion.

is hereinafter referred to as the "best" variant (for the problem at hand).

In the following subsections, the influence of MPDATA algorithm variant choice on the resultant spectrum width and on the computational cost is analysed using the example simulation set-up used above (i.e. in all figures except Fig. 2; see Sect. 2.2 for test case definition). Analysis of the scheme solution convergence with changing resolution and Courant number is presented in Appendix A.

## 2.11 Quantification of numerical broadening

The relative dispersion, defined as the ratio of standard deviation $\sigma$ to the mean $\mu$ of the spectrum, is a parameter commonly used to describe the width of the spectrum (e.g. Chandrakar et al., 2018).

**Table 1.** Relative dispersion of the discretised (using grid set-up as in Fig. 1) analytical solution taken for five selected times.

| Variant | $d_{\text{analytical}}$ |
|---|---|
| $d(M = 1\,\text{g}\,\text{kg}^{-1})$ | 0.357 |
| $d(M = 2\,\text{g}\,\text{kg}^{-1})$ | 0.202 |
| $d(M = 4\,\text{g}\,\text{kg}^{-1})$ | 0.126 |
| $d(M = 6\,\text{g}\,\text{kg}^{-1})$ | 0.097 |
| $d(M = 8\,\text{g}\,\text{kg}^{-1})$ | 0.080 |
| $d(M = 10\,\text{g}\,\text{kg}^{-1})$ | 0.069 |

The calculated dispersion ratio over all bins takes the following form:

$$d = \frac{\sqrt{\frac{1}{N}\sum_i m_i^{(l=2)} - \left(\frac{1}{N}\sum_i m_i^{(l=1)}\right)^2}}{\frac{1}{N}\sum_i m_i^{(l=1)}}, \tag{25}$$

where $m_i$ is defined in Eq. (8) and $N$ is the conserved total number of particles (equal to $\sum_i m_i^{(l=0)}$). To quantify the effect of the numerical diffusion on the broadening of the spectrum, the following parameter is introduced based on the numerical and analytical solutions (hereinafter reported in percentages):

$$R_d = d_{\text{numerical}}/d_{\text{analytical}} - 1. \tag{26}$$

Table 1 depicts the gradual narrowing of the spectrum under undisturbed adiabatic growth. The left-hand panel in Fig. 9 provides values of the $R_d$ parameter evaluated at six selected times corresponding to $M = 1, 2, 4, 6, 8, 10\,\text{g}\,\text{kg}^{-1}$. Although numerical broadening is inherent to all employed schemes and grows in time for all considered variants, the scale of the effect is significantly reduced when using MPDATA. In particular, a 10-fold decrease in numerical broadening as quantified using $R_d$ is observed comparing upwind and the "best" variant considered herein.

### 2.12 Notes on conservativeness

Due to the formulation of the problem as number conservation, and discretisation of the evolution equation using fixed bins, even though the numerical scheme is conservative (up to subtle limitations outlined below), evaluation of other statistical moments of the evolved spectrum from the number density introduces an inherent discrepancy from the analytical results (for a discussion on multi-moment formulation of the problem, see Liu et al., 1997).

In order to quantify the discrepancy in the total mass between the discretised analytical solution and the numerically integrated spectrum, the following ratio is defined using the moment evaluation formula (Eq. 8):

$$R_m = M^{(\text{numerical})}/M^{(\text{analytical})} - 1 = \tag{35}$$

$$= \frac{\sum_i m_i^{(l=3,\,\text{numeric})}}{\sum_i m_i^{(l=3,\,\text{analytical})}} - 1. \tag{27}$$

The right-hand panel in Fig. 9 depicts the values of the above-defined ratio computed for spectra obtained with different variants of MPDATA discussed herein. The departures from analytically derived values are largest for the upwind scheme (up to ca. 5 %) and oscillate around 0 with an amplitude of the order of 1 % for most of the MPDATA solutions.

The consequences of mass conservation inaccuracies in the fixed-bin particle-size spectrum representation may not be as severe as in, for example, a dynamical core responsible for the transport of conserved scalar fields. The outlined discrepancies may be dealt with by calculating the change in mass during a time step from condensation, then using it in vapour and latent heat budget calculations so the total mass and energy in the modelled system are conserved.

The embraced algorithm (Eqs. 3–4) is conservative (up to numerical precision) for $G = 1$. However, the formulation of the donor cell scheme $\psi^{n+1} = \psi^n + G_i^{-1}\left(F_{i-1/2} + F_{i+\frac{1}{2}}\right)$ on the staggered grid with $G \neq 1$, for example due to employment of non-identity coordinate transformations, implies that even though the influx and outflux across boundary of adjacent cells is equal, discretisation of $G_i$ at cell centres limits the level of accuracy in number conservation. For further discussion, see Sect. 3 in Smolarkiewicz and Rasch (1991).

The total number of particles in the system may diverge from the analytical expected value even for the initial condition depending on the employed discretisation approach. In the present work, the probability density function is sampled at cell centres effectively assuming piecewise-constant number density function. An alternative approach is to discretise the initial probabilities by assigning to $\psi_i$ the values of $(\phi_{i+\frac{1}{2}} - \phi_{i-1/2})/(r_{i+\frac{1}{2}} - r_{i-1/2})$ where $\phi$ is the cumulative distribution.

### 2.13 Computational cost

Table 2 includes an assessment of the relative computational cost of the explored variants of MPDATA. The performance was estimated by repeated measurements of the wall time and selecting the minimal value as representative. Values are reported as multiplicities of the upwind execution time. Simulations were performed using the mass-doubling grid.

The table includes, where available, analogous figures reported in earlier studies on MPDATA (see caption for comments on the dimensionality of the employed cases, as it differs and thus does not warrant direct comparison). Among notable traits is the decrease in computational cost when enabling the infinite gauge option that is associated with a reduced number of terms in both the flux function as well

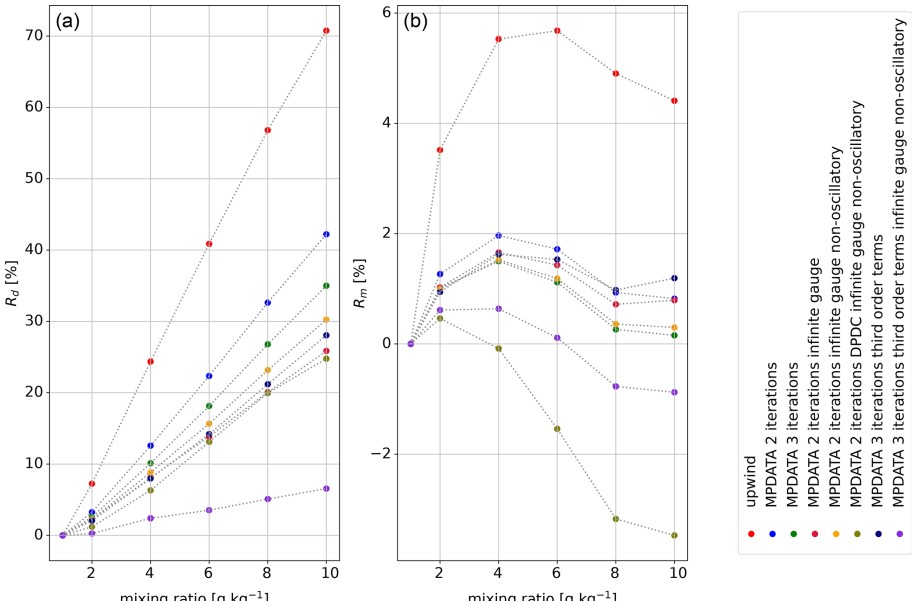

**Figure 9.** Panel **(a)** summarises values of the numerical-to-analytical spectral width ratio $R_d = d_{numerical}/d_{analytical} - 1$ (expressed as a percentage) computed for simulations using different discussed variants of MPDATA and plotted as a function of increasing mixing ratio (i.e. each simulation is depicted with a set of line-connected points corresponding to selected time steps); see Sect. 2.11. Panel **(b)** presents analogous analysis for $R_m$; see Sect. 2.12 for discussion. Note: $R_m = R_d = 0$ corresponds to a perfect match with the analytical solution. TS4

as in the antidiffusive velocity formulation (see Sect. 2.5 in Hill, 2011, and Sect. 2.5–2.6 herein). The "best" variant is roughly 10 times more costly than the upwind scheme for the set-up studied herein and the employed implementation. Among studies of bin microphysics schemes, analogous measures were reported in Liu et al. (1997) where the variational method presented there was reported to take 3.1 times longer to execute than the first-order upwind solution, and in Onishi et al. (2010) where the studied semi-Lagrangian scheme was reported to be characterised by a computational cost over 4 times higher than the upwind scheme (see Table 4 therein). In the latter case, a direct comparison is hindered by significantly different stability constraints on the time step.

Although the discussed problem is one-dimensional, a computationally efficient and an accurate solution is essential, as it typically needs to be solved at every time step and grid point of a three-dimensional cloud model. While the reported upwind-normalised wall times give a rough estimation of the cost increase associated with a particular MPDATA option, the actual footprint on a complex simulation system will depend on numerous implementation details including parallelisation strategy.

**Table 2.** Wall times normalised with respect to the upwind solution compared to data reported in four earlier works: S83 denotes Smolarkiewicz (1983) (two-dimensional problem), SS05 corresponds to Smolarkiewicz and Szmelter (2005) (three-dimensional, unstructured grid), SR91 denotes Smolarkiewicz and Rasch (1991), and MSS00 corresponds to Margolin et al. (2000) (both reported for two-dimensional problems).

| Variant | | S83 | SS05 | SR91 | MSS00 |
|---|---|---|---|---|---|
| Upwind | 1.0 | 1.0 | 1.0 | 1.0 | 1.0 |
| 2 iterations | 2.5 | 2.9 | 4.3 | 5.4 | 3.7 |
| 2 iterations, iga | 2.2 | – | 1.9 | – | – |
| 2 iterations, iga, non–osc | 5.9 | – | 3.9 | – | – |
| DPDC, iga, non–osc | 6.2 | – | – | – | – |
| 3 iterations | 5.7 | 5 | – | 9.8 | – |
| 3 iterations, tot | 4.1 | – | – | 19 | – |
| 3 iterations, tot, iga, non–osc | 11 | – | – | – | – |

## 3   Spectral-spatial advection with MPDATA (single-column test case)

### 3.1   Problem statement

In multidimensional simulations in which the considered particle number density field is not only a function of time and particle size, but also of spatial coordinates, there are several additional points to consider applying MPDATA to the problem.

First, in the context of atmospheric cloud simulations, owing to the stratification of the atmosphere, the usual prac-

tice is to reformulate the conservation problem in terms of specific number concentration being defined as the number of particles $n_p$ (cf. Eq. 1) divided by the mass of air (commonly the dry air) effectively resulting in including the (dry) air density in the $G$ factor (see Eqs. 2–3). This is motivated by atmospheric stratification associated with the presence of a vertical air density gradient. In a non-divergent stratified flow, the specific number concentration (summed across all particle-size categories) is not modified by advection along the vertical dimension. On the other hand, particle volume concentration (as opposed to specific number concentration) would vary due to variable density of air (i.e. expansion of air along the vertical coordinate). Note that in Eq. (3) it is further assumed that the $G$ factor does not vary in time.

Second, even with a single spatial dimension (single-column set-up), the coupled size-spectral–spatial advection problem is two-dimensional. This is where the inherent multidimensionality of MPDATA (the "M" in MPDATA) requires further attention. The one-dimensional antidiffusive formulæ discussed in Sect. 2.4–2.9 need to be augmented with additional terms representing cross-dimensional contributions to the numerical diffusion. For an introduction, see for example Sect. 2.2 in Smolarkiewicz and Margolin (1998), for original derivation see Smolarkiewicz (1984), and for a recent work discussing the interpretation of all terms in the antidiffusive velocity formulæ, including cross-dimensional terms, see Waruszewski et al. (2018).

Third, in any practical application where the drop size evolution is coupled with the water vapour budget (and hence with supersaturation evolution), it is essential to evaluate the total change in mass of liquid water due to condensation which is then to be used to define the source term of the water vapour field (and in latent heat budget representation). It is worth noting that knowing the difference of values at $n + 1$ and at $n$ time steps of the advected specific number concentration field is not sufficient to evaluate the vapour sink–source term. This is because the fluxes across the size-spectral dimension only need to be accounted for (note that the fluxes in all MPDATA iterations need to be summed up).

Several recent papers are highlighting the need for scrutiny when comes to the interplay of size-spectral and spatial advection and the associated numerical broadening (Morrison et al., 2018; Hernandéz Pardo et al., 2020; Lee et al., 2021). In the following subsection, a set of single-column simulations is presented and discussed depicting the performance of MPDATA in a size-spectral–spatial advection problem coupled with vapour advection and supersaturation budget. The simulations are performed using a commonly employed MPDATA setting with only the non-oscillatory option enabled, and the discussion is focused on the sensitivity of the results to spatial, spectral and temporal resolution, as well as to the effect of performing one or two corrective passes of MPDATA (two or three iterations, respectively).

## 3.2    Test case definition

The test set-up is based on the single-column KiD warm case introduced in Shipway and Hill (2012). This prescribed-flow framework has been further used, e.g. in Field et al. (2012) (mixed-phase scenario), in Gettelman and Morrison (2015) (both pure-ice, mixed-phase and warm-rain scenarios), and in the microphysics-model intercomparison study (warm rain scenario). CE1 Here, condensation is the only microphysical process considered.

The simulated 3.2 km high column of air is described by the following:

- a constant-in-time piecewise-linear potential temperature profile (297.9 K from the ground to the level of 740 m, linearly decreasing down to 312.66 K at 3260 m);

- constant-in-time hydrostatic pressure and density profiles computed assuming surface pressure of 1007 hPa;

- a piece-wise linear initial vapour mixing ratio profile (15 g kg$^{-1}$ at ground, 13.8 g kg$^{-1}$ at 740 m and 2.4 g kg$^{-1}$ at 3260 m); and

- a constant-in-space but time-dependent vertical momentum profile defining the vertical component of the advector field $GC_z$ TS6 as in Eq. (28): TS7

$$ GC_z(z,t) = \rho_d(z) \frac{\Delta t}{\Delta z} w_1 \sin(\pi t/t_1)(1 - H(t - t_1)), \quad (28) $$

where $H$ is the Heaviside step function, $w$ is the vertical velocity, $w_1 = 2.5 \, \mathrm{m \, s^{-1}}$ TS8, $\rho_d(z)$ is the hydrostatic dry density profile and $t_1 = 600$ s.

Note that the vertical velocity thus differs from the original KiD set-up where $w$ is held constant across the column (rather than constant momentum density as done herein). This change is motivated by the aim of maintaining the non-divergent flow field condition.

The advection is thus solved for two scalar fields: (i) a one-dimensional water vapour mixing ratio field representing the vertical distribution of mass of vapour per mass of dry air and (ii) a two-dimensional field representing vertical and spectral variability of liquid particle specific concentration (number of particles per mass of dry air). The spectral coordinate is set to particle radius ($p = r$) and the bins are laid out uniformly ($x = r$) over a range of 1 to 20.2 μm. It is worth noting that this results in the size-spectral component of the advection velocity being divergent (while the vertical component is non-divergent).

The initial condition does not feature supersaturation anywhere in the domain. The upward advection of water vapour causes supersaturation to occur and trigger condensation. The size-spectral velocity is defined as in the box-model test case (cf. Eq. 5) but with supersaturation being time-dependent and derived from the values of vapour mixing ratio, temperature and pressure at a given level. Note that the

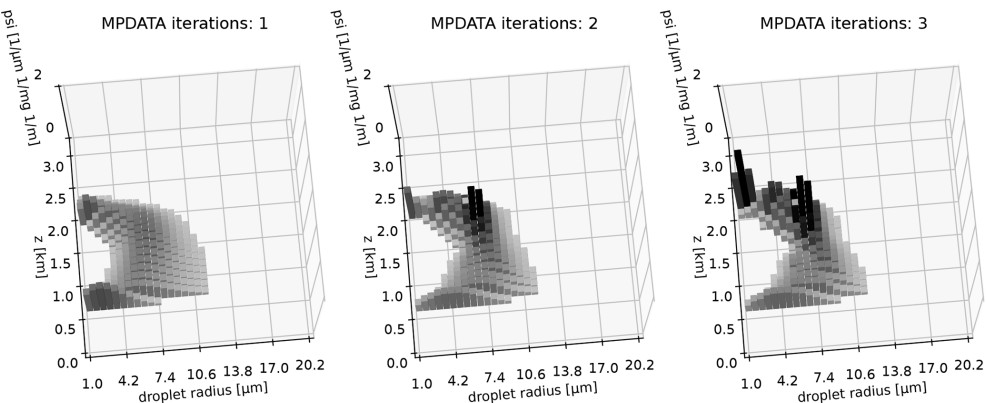

**Figure 10.** Snapshots of the advected two-dimensional liquid water field at $t = t_1 = 600\,\mathrm{s}$ for simulations with different number of MPDATA iterations (see text for details). TS5

temperature profile is constant in time and the test case does not feature representation of latent heat release effects, only the ambient air and particle vapour budget is accounted for by subtracting the amount of condensed water from the vapour field in each time step, before performing the subsequent step of advection on the vapour mixing ratio field.

The domain is initially void of liquid water and the only source of it is through the boundary condition in the spectral dimension specified as follows:

$$\psi_{-1} = \max\left(0, N_{\mathrm{CCN}} - \sum_i \psi\right), \tag{29}$$

with $i = -1$ denoting the halo grid cell at the left edge of the spectral domain on a given vertical level (the summation spans all bins at a given level excluding the halo grid cells). The flux across the domain boundary in the spectral dimension represents cloud droplet activation. Through Eq. (5), the flux is dependent on the supersaturation at a given level, and on the $N_{\mathrm{CCN}}$ parameter representing a maximal number of activated droplets (per unit mass of dry air). In the performed simulations, $N_{\mathrm{CCN}}$ was set to $500\,\mathrm{mg}^{-1}$. For discussion of other ways to represent activation in bin microphysics models, see for example Grabowski et al. (2011).

The simulations cover a time period of 15 min out of which the first 10 min (until $t_1 = 600\,\mathrm{s}$) involve non-zero vertical velocity (cf. Eq. 28). Several temporal, spatial and spectral resolutions are tested with the following settings hereinafter referred to as the base resolution case: $32 \times 32$ grid with a vertical grid step $\Delta z = 100\,\mathrm{m}$, size-spectral grid step $\Delta r = 0.6\,\mathrm{\mu m}$ and time step $\Delta t = 0.25\,\mathrm{s}$.

### 3.3 Discussion of results

Figure 10 depicts qualitatively how MPDATA performs with the single-column simulation (base resolution case) depending on the number of MPDATA iterations employed. The two-dimensional liquid water mixing ratio grid is rendered with shaded histogram bars. The vertical axis corresponds to the advected quantity: spatio-spectral number density divided by the dry density of air. Histogram bars with values of less than $1\,\%$ TS9 of the vertical axis range ( TS10 $1\,\% \times 2\,\mathrm{m}^{-1}\,\mathrm{mg}^{-1}\,\mathrm{\mu m}^{-1}$) are not plotted for clarity. Presented plots are aimed at intuitively portraying the model state and the extent to which the introduction of subsequent MPDATA corrective iterations reduces spectrum broadening. Note that besides the depicted liquid water mixing ratio, the model state consists as well of a one-dimensional vapour mixing ratio vector (not shown).

In Fig. 11, the base resolution case is depicted with plots constructed following the original methodology from Shipway and Hill (2012) (as in Fig. 1 therein). The greyscale maps depict the evolution in time and vertical dimension of water vapour mixing ratio $q_1$, supersaturation $S$ and the droplet spectrum relative dispersion $d$. The adjacent profile plots depict the vertical variability of the mapped quantity at four selected times.

Notwithstanding the highly idealised and simplified modelling framework employed herein, one may attempt a comparison with profiles obtained from both in situ aircraft measurements (Arabas et al., 2009, profiles of $d$ in Fig. 1 therein) and detailed three-dimensional simulations (Arabas and Shima, 2013, profiles of $S$ and liquid water content in Figs. 2–4 therein) inspired by the same RICO field campaign (Rauber et al., 2007) as the single-column set-up of Shipway and Hill (2012). The comparison confirms that the chosen test case covers the parameter space relevant to the studied problem. Resemblance remains, at most, qualitative, as expected given the stark simplicity of the KiD framework.

Interestingly, the parabolic vertical profile of the relative dispersion obtained herein was also reported in Lu and Seinfeld (2006) for bin-microphysics three-dimensional simulations of marine stratocumulus. In the discussion of Figs. 2, 3 and 6 therein, it was hypothesised that the parabolic shape is a signature of entrainment as well as updraft–downdraft

**https://doi.org/10.5194/gmd-15-1-2022** Geosci. Model Dev., 15, 1–21, 2022

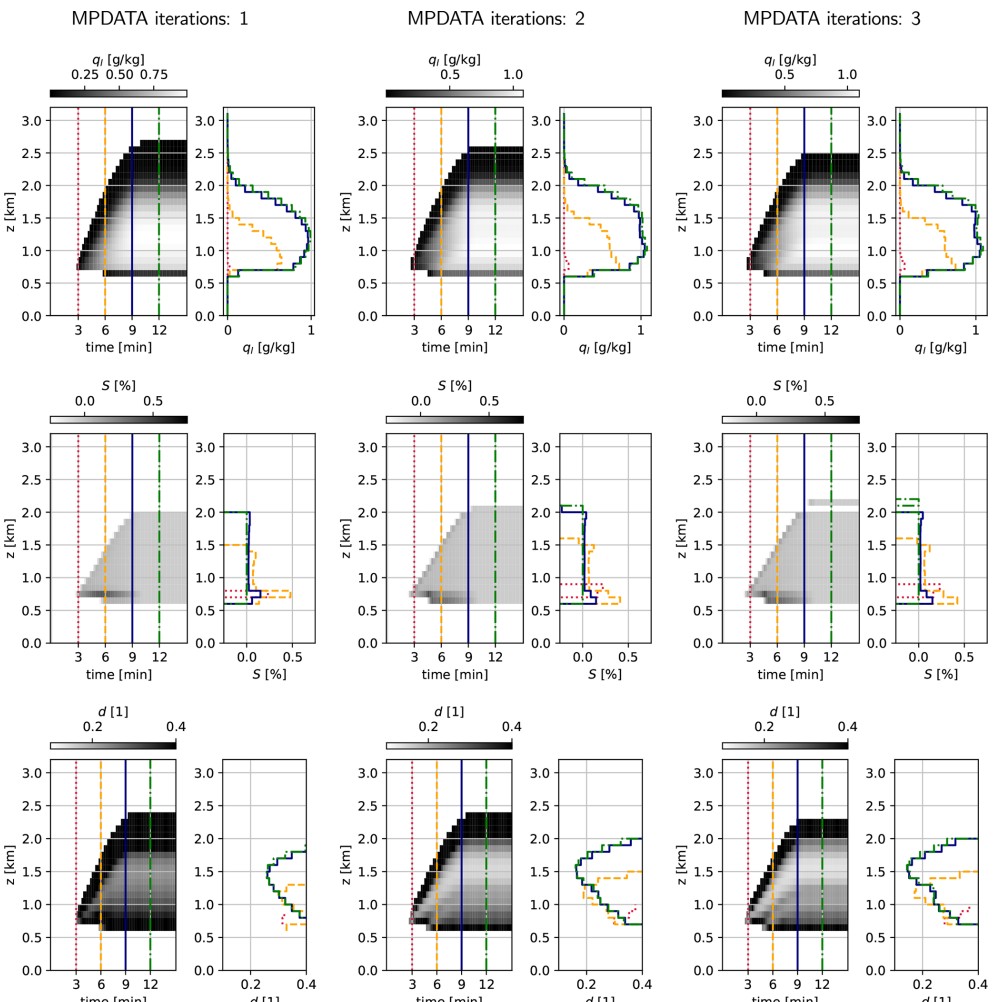

**Figure 11.** Single-column simulations depicted with three selected variables: liquid water mixing ratio $q_1$ (top row), supersaturation $S$ (middle row) and relative dispersion $d$ (bottom row); for three settings of the iteration count in MPDATA (one iteration corresponding to the basic upwind scheme, left-hand column). Each of nine datasets (three iteration settings, three variables) is plotted with a greyscale time vs. altitude map (left-hand panels with the colour scale above) and a set of four profiles (right-hand panels). Profiles are plotted for $t = 3$ min (dotted, red), 6 min (dashed, orange), 9 min (solid, navy) and 12 min (dash-dot, green), with vertical lines of corresponding line style plotted at given times in the left-hand panels. For plotting, the model state is resampled by averaging in the time dimension to reduce the number of plotted steps by a factor of 50 (from 3600 down to 72).

interactions, none of which are represented in the kinematic framework employed herein.

The liquid water profiles depicted in the top row of Fig. 11 reveal that the cloud structure developed within the first ca. 9 min of the simulation is later maintained, with the profiles at $t = 9$ min and $t = 12$ min being virtually indistinguishable. Middle row plots of supersaturation profiles depict that the considered simulation set-up enables the user to capture the characteristic supersaturation maximum just above cloud base. Furthermore, it is evident that the corrective iterations of MPDATA influence the maximal supersaturation values. It is worth noting that this results in different time step (Courant number) constraints depending on the number of iterations used because the spectral velocity is a function of supersaturation.

There is a cloud-top activation feature hinted in all three panels in Fig. 10 as well as indirectly in the supersaturation profiles in Fig. 11. The representation of activation above cloud base is sensitive to both numerical details of vapour and heat transport reflected in the diagnosed supersaturation, as well as to the assumptions behind the activation formulation itself (see for example the discussion of Figs. 2 and 6 in Slawinska et al., 2012, and references therein). Given the simplified treatment of activation defined by Eq. (29), together with the unphysical assumption of constant temperature profile, no physical interpretation of this feature is warranted. Yet, it is worth noting that, consistent with the dif-

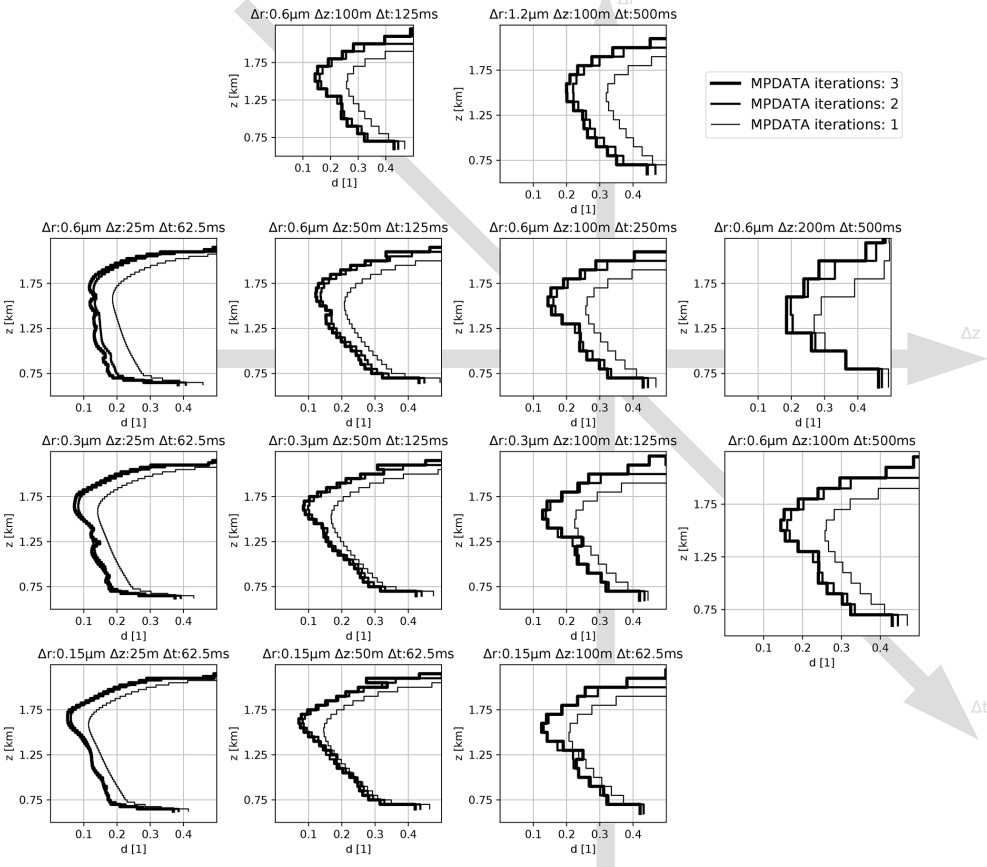

**Figure 12.** Profiles of relative dispersion $d$ for a set of temporal, spatial and spectral resolution settings ($\Delta r$, $\Delta z$ and $\Delta t$ values given in labels above each plot). Each panel depicts results for three different MPDATA iteration counts (one iteration corresponding to the basic upwind scheme). Profiles plotted for $t = t_1 = 10$ min.

ferences in supersaturation values between upwind and MP-DATA solutions, the cloud-top activation is least noticeable in Fig. 10 in the case of the upwind solution.

The bottom row in Fig. 11 depicts the relative disper-
5 sion defined and computed as in Sect. 2.11 (discarding levels where the total droplet number mixing ratio summed over all bins on a level is below 5 % of $N_{\mathrm{CCN}}$). Narrowing of the spectrum with height below $z = 1.5$ km revealed by decreasing values of $d$ is a robust feature. Minimum values of $d$ for
a given profile vary visibly depending on the number of MP-DATA iterations employed.

To provide insight into the sensitivity of the results to temporal, spatial and spectral resolution, Fig. 12 presents the relative dispersion profiles at $t = t_1 = 10$ min for several resolu-
15 tion settings. In the background of the figure, there are three axes plotted pointing in the directions in which the figure panels can be explored to reveal the dependence on the vertical spatial spacing $\Delta z$ (left-to-right), the spectral spacing $\Delta r$ (bottom-to-top) and the time step (back-to-foreground).
The base resolution case is plotted at the intersection of the axes. Note that besides the back-to-foreground sequence of

plots where all but the time step settings is kept equal, the time step also varies with the grid settings to fulfil scheme stability constraint.

The dependence on the temporal resolution, as gauged by
25 comparing the base resolution case with cases in which the time step is halved ($\Delta t = 125$ ms; background) and is doubled ($\Delta t = 500$ s; foreground), remains barely observable. This is in general agreement with Morrison et al. (2018) and Hernandéz Pardo et al. (2020) where the dependence on the
30 time step is shown to be much smaller than on the spatial or on the spectral resolution.

The dependence on the spectral resolution is clearly manifested at the lowest spectral resolution where the minimum spectral dispersion $d$ along a profile drops by ca. 0.1 when
decreasing $\Delta r = 1.2$ μm down to $\Delta r = 0.3$ μm. Little further change can be observed by refining the resolution down to $\Delta r = 0.15$ μm. Focusing on the minimum values of $d$ for a given profile, in general, the lower the spectral resolution, the more profound the effect of introducing corrective iterations
of MPDATA. In most cases, applying even a single corrective step (i.e. two iterations) results in halving of the minimal

values of $d$ as compared to the upwind solution (i.e. one iteration).

The spatial resolution setting $\Delta z$ significantly alters the results, particularly near the cloud base. The values of $d$ at the lower half of the presented profile (i.e. ca. below $z = 1$ km) drop from over 0.3 down to around 0.1 TS11 when refining the resolution from $\Delta z = 200$ m down to $\Delta z = 25$ m.

## 4   Conclusions

The study focuses on the MPDATA family of numerical schemes and its application to the size-spectral as well as spatio-spectral transport problem arising in models of condensational growth of cloud droplets. MPDATA iteratively applies the upwind algorithm, first with the physical velocity and subsequently using antidiffusive velocities. As a result, the algorithm is characterised by reduced numerical diffusion compared with upwind solutions, while maintaining conservativeness and positive-definiteness.

In literature, the derivations of different MPDATA variants are spread across numerous research papers published across almost four decades, and in most cases focused on multidimensional hydrodynamics applications. It is the aim of this study to highlight the developments that followed the original formulation of the algorithm, and to highlight their applicability to the problem of bin microphysics. To this end, it was shown that the combination of such features of MPDATA as the infinite-gauge, non-oscillatory and third-order-term options, together with the application of multiple corrective iterations offer a robust scheme that grossly outperforms the almost quadragenarian basic MPDATA.

In the case of the single-column test case, discussed simulations feature coupling between droplet growth and supersaturation evolution. The embraced measure of spectrum width, the cloud droplet spectrum relative dispersion, is influenced by numerical diffusion pertinent to both spectral and vertical advection. Focusing on the levels corresponding to the region of maximal liquid water content (ca. between $z = 1$ and 2 km for the case considered), application of even a single corrective iteration of MPDATA robustly reduces (in most cases more than halves) the spectral width. In agreement with conclusions drawn from single-column simulations in Morrison et al. (2018) and Lee et al. (2021), within the range of explored grid settings, the vertical resolution has the most profound effect on the overall characteristics of the spectrum width profile as it significantly influences the just-above-cloud-base evolution of the spectral width.

## Appendix A:  Convergence analysis

To assess the spatial and temporal convergence of the numerical solutions presented above, a convergence test originating from Smolarkiewicz and Grabowski (1990) is used. For the analysis the following truncation-error $L^2$ measure is used

(e.g. Smolarkiewicz, 1984):

$$\mathrm{Err}_{L2} = \frac{1}{T} \sqrt{\sum_i \left( \psi_i^{\mathrm{numerical}} - \psi_i^{\mathrm{analytical}} \right)^2 / nx}. \qquad (A1)$$

As a side note, it is worth pointing out that for the chosen coordinates $\left( p = r^2, x = r^2 \right)$, the coordinate transformation term is equal to the identity, so there is no need for including the $G$ factor into the computed error measures. In the general case, convergence will depend on the grid choice and to account for that one may use a modified measure as given in Smolarkiewicz and Rasch (1991, Eq. 24).

To explore the convergence, the error measures are computed for seven different linearly spaced values of $C$ between 0.05 and 0.95, and $nx \in \left\{ 2^7, 2^8, 2^9, 2^{10}, 2^{11}, 2^{12}, 2^{13}, 2^{14} \right\}$ resulting in 56 simulations for each presented combination of options.

As proposed in Smolarkiewicz and Grabowski (1990), visualisation of the results is carried out on polar plots with radius $\rho$ and angle $\phi$ coordinates defined as follows:

$$\rho = \ln_2 \left( \frac{1}{nx} \right) + \mathrm{const}, \quad \phi = C \frac{\pi}{2}, \qquad (A2)$$

where $\rho$ was shifted by a constant so that the highest resolution grid corresponds to $\rho = 1$.

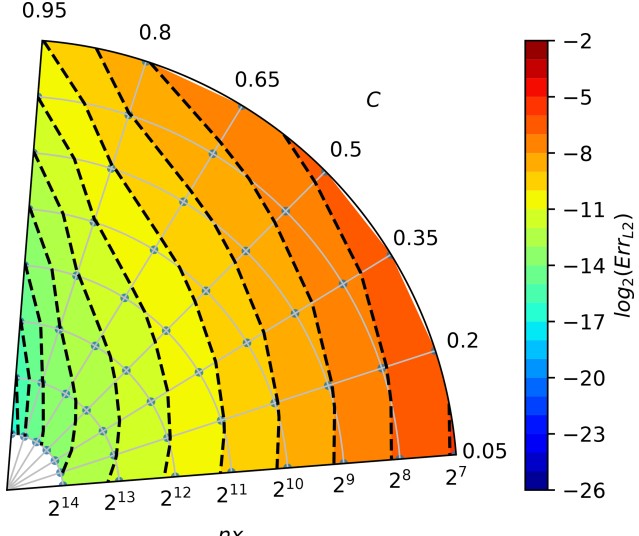

**Figure A1.** Convergence plot for the upwind scheme (cf. Fig. 1). Angle in the polar plot corresponds to the Courant number $C$, and the distance from origin denotes the number of grid boxes $nx$; see Eq. (A2). Grey dots indicate data point locations – parameter values for which computations were made. Colours and isolines depict the error measure values (interpolated from the data point locations), see Eq. (A1).

Figures A1–A8 depict the convergence rates and are intended for comparison with analogously constructed plots in Smolarkiewicz and Grabowski (1990, Figs. 2–3), Margolin

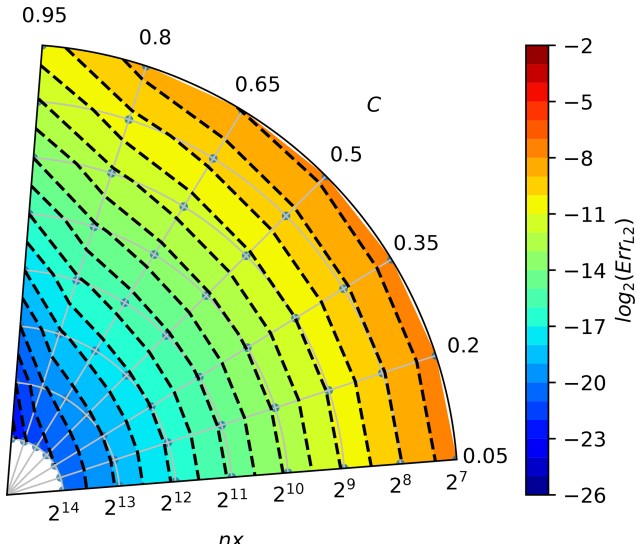

**Figure A2.** Convergence plot for basic two-pass MPDATA (cf. Fig. 3). See caption of Fig. A1 for the description of plot elements.

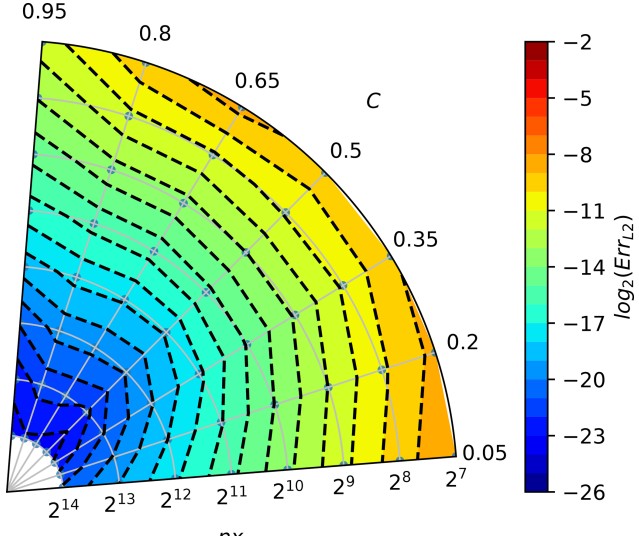

**Figure A4.** Convergence plot for the infinite gauge non-oscillatory variant of MPDATA (cf. Fig. 5). See caption of Fig. A1 for the description of plot elements.

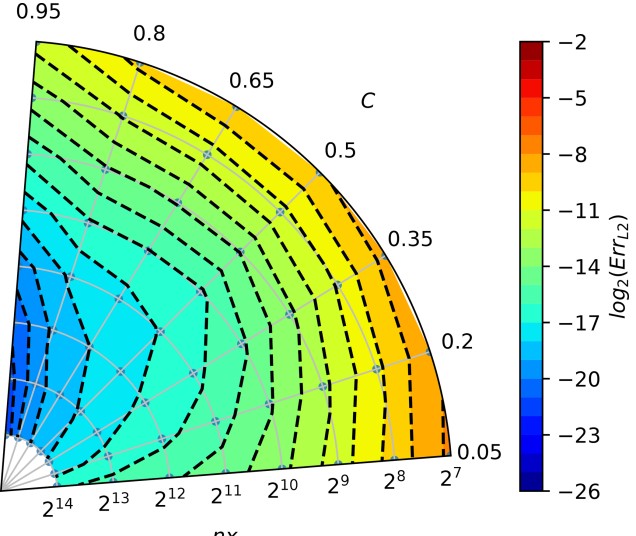

**Figure A3.** Convergence plot for the infinite gauge MPDATA (cf. Fig. 4). See caption of Fig. A1 for the description of plot elements.

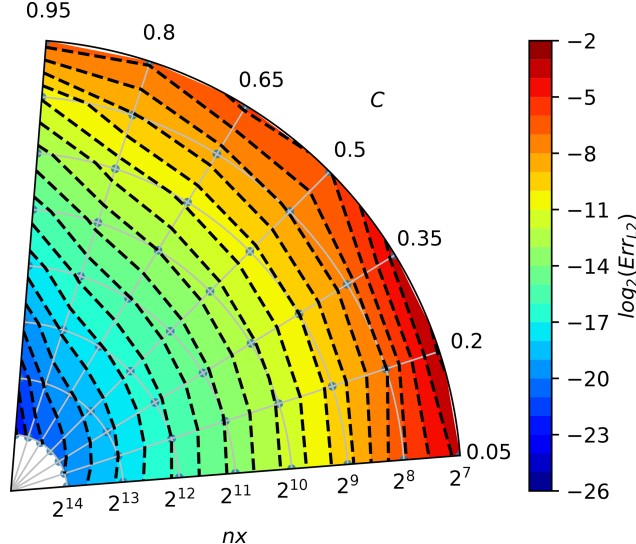

**Figure A5.** Convergence plot for the DPDC variant with infinite gauge and non-oscillatory corrections (cf. Fig. 6). See caption of Fig. A1 for the description of plot elements.

and Smolarkiewicz (1998, Figs. 8.1–8.2) and Jaruga et al. (2015, Figs. 10–11) TS12.

The chosen colour increments correspond to the error reduction by a factor of 2, and the warmer the colour, the larger the error. The small grey points behind the isolines represent points for which the error value was calculated. When moving along the lines of constant Courant number, increasing the space and time discretisation, the number of crossed dashed isolines indicate the order of convergence. For the considered problem, it can be seen that the upwind scheme (Fig. A1) has a convergence of the first order (one isoline is crossed when spatial discretisation increases by one order); the MPDATA scheme (Fig. A2) is of the second order, and MPDATA with 3 iterations (Fig. A6) is of the third order.

Moreover, the shape of the dashed isolines tells the dependency of the solution accuracy on the Courant number. When these are isotropic (truncation error being independent of polar angle), the solution is independent of the Courant number.

It is worth noting that in Figs. A3 and A4, a groove of the third-order convergence rate is evident around $\phi = \frac{\pi}{4}$, normally characteristic for MPDATA with three or more passes.

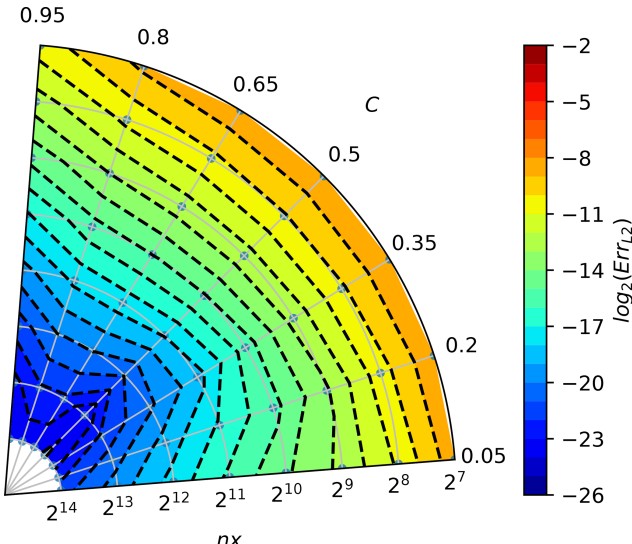

**Figure A6.** Convergence plot for the three-pass MPDATA (cf. Fig. 3). See caption of Fig. A1 for the description of plot elements.

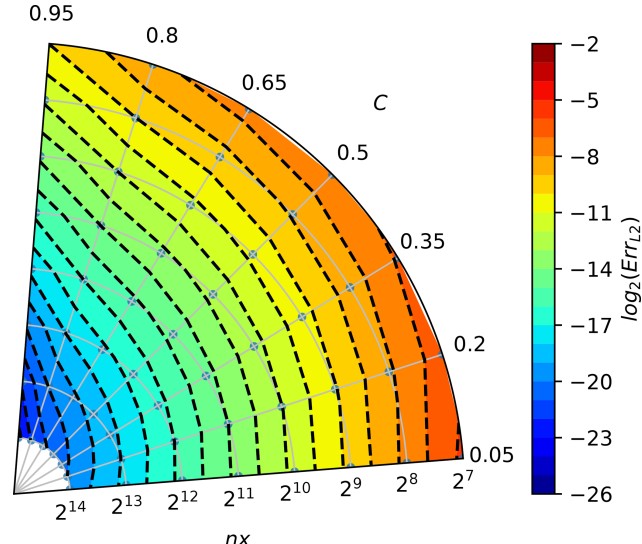

**Figure A8.** Convergence plot for the three-pass infinite gauge non-oscillatory MPDATA with third-order term corrections (cf. Fig. 8). See caption of Fig. A1 for the description of plot elements.

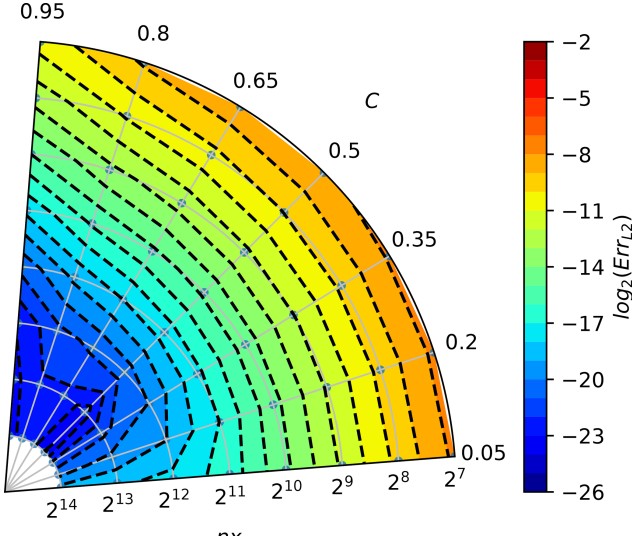

**Figure A7.** Convergence plot for the three-pass MPDATA with third-order terms (cf. Fig. 7). See caption of Fig. A1 for the description of plot elements.

*Code availability.* Calculations presented in the paper were performed using Python with a new open-source implementation of MPDATA: PyMPDATA (Bartman et al., 2022a).TS13 All of presented figures and tables can be recreated using Jupyter notebooks included in the PyMPDATA-examples package. Both PyMPDATA and PyMPDATA-examples are licensed under the GNU General Public License 3.0 and are available on the PyPI.org Python package repository. Releases are being additionally archived on zenodo.org. The DOI links for versions 1.0 used in this study are https://doi.org/10.5281/zenodo.6329303 (PyMPDATA 1.0, Bartman et al., 2022a) and https://doi.org/10.5281/zenodo.6471494 (PyMPDATA-examples 1.0.1) (Bartman et al., 2022c).TS14

*Data availability.* No data sets were used in this article.TS15

*Author contributions.* The idea of the study originated in discussions between SA, SU and MAO. MAO led the work, and a preliminary version of a significant part of the presented material constituted his MSc thesis prepared under the mentorship of SA. PB architected the key components of the PyMPDATA package. JB contributed the DPDC variant of MPDATA to PyMPDATA. MB participated in composing the paper and devising the result analysis workflow. All authors contributed to the final form of the text.

*Competing interests.* At least one of the (co-)authors is a member of the editorial board of *Geoscientific Model Development*. The peer-review process was guided by an independent editor, and the authors also have no other competing interests to declare.

When second-order truncation error is sufficiently reduced, the third-order error, proportional to $(1 - 3C + 2C^2)$ as can be seen in Eq. (23), dominates but vanishes for $C = 0.5$, thus resulting in the existence of the groove.

The convergence test results for the three-pass MPDATA with infinite gauge, non-oscillatory and third-order terms options enabled (Fig. A8) are consistent with results depicted in Fig. A7, although the order of convergence is reduced due to the employment of non-oscillatory option.

*Acknowledgements.* Comments from Wojciech Grabowski, Adrian Hill, Hugh Morrison, Andrzej Odrzywołek, Piotr Smolarkiewicz and Maciej Waruszewski as well as paper reviews by Josef Schröttle and three anonymous reviewers helped to improve the article. TS16

*Financial support.* The project was carried out within the POWROTY/REINTEGRATION programme of the Foundation for Polish Science (https://www.fnp.org.pl/, last access: April 2022) co-financed by the European Union under the European Regional Development Fund (POIR.04.04.00-00-5E1C/18-00). Manuel Baumgartner acknowledges support by the Deutsche Forschungsgemeinschaft (DFG) within the Transregional Collaborative Research Centre TRR165 Waves to Weather (https://www.wavestoweather.de/, last access: April 2022) project Z2. Simon Unterstrasser acknowledges support through the DLR-internal research group H2CONTRAIL. The publication was partially funded through the "Excellence Initiative – Research University" programme at the Jagiellonian University.

*Review statement.* This paper was edited by Juan Antonio Añel and reviewed by Josef Schröttle and three anonymous referees.

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

## Remarks from the language copy-editor

## Remarks from the typesetter

**TS2** Thank you for your feedback. Please note that vector graphics (*.eps, *.ps) cannot be included in the PDFLaTeX for technical reasons. In addition, *.pdf figures cannot be included in the PDFLaTeX since certain fonts or other content cannot be embedded and such content would then not show up in some browsers or *.pdf viewers. As a result, affected figures might appear incomplete to some readers. Therefore, we only include *.png and *.jpg figures in the article *.pdf. However, since we also publish all articles in full-text HTML, we will provide your vector graphics as high-resolution figures so that readers are able to download and enlarge the figures for re-use (see e.g. https://gmd.copernicus.org/articles/15/2505/2022/). Please note that this high-resolution download is only possible if your figure has the Creative Commons Attribution 4.0 License (CC BY) applied. This is the case for the figures compiled by you or your co-authors. If you cite a figure from another paper that is not distributed under the Creative Commons Attribution License, the figure is identified as protected and the download link will be hidden.