# Peer review of "On numerical broadening of particle-size spectra: a condensational growth study using PyMPDATA 1.0"

_Geoscientific Model Development, 2020_

## Referee Comment (RC1)

Review for **Geoscientific Model Development** (Feb. 24$^{th}$ 2021)

On numerical broadening of particle size spectra: a condensational growth study using PyMPDATA 1.0

This study introduces a new Python library for advection of geophysical flows with the MPDATA scheme. More specifically, it concentrates on the broadening of cloud droplet distributions due to advection and compares those distributions to analytically derived functions. It is based on previous work by the authors in a C++ library and numerous studies that have been conducted since the 1980s up to now. The paper is clearly written in the introduction, the methodology, and results sections. The reader would profit from a more fluent overview of the background literature in this work (in **section 1.2**), as well as a brief motivation why this is such important work especially in the context of clouds. A short suggestion of how to incorporate this is given by the reviewer. Also, a brief description of the software for interested users would be very helpful. Overall, this work is unique in its focus on comparing 2-3 advection schemes in the context of cloud dynamics and could be a basis for many future applications after major revision.

First general comments from my side:

- To give a brief motivation, why your work is so important, I would refer to clouds in climate research, e.g.: Climate goals and computing the future of clouds, *Nature Climate Change* **volume** 7, pages3–5 (2017) or a more recent publication

- How do your simulations evolve in time? Besides the cloud distributions you show in Fig. 1ff, I would like to see Hovmoeller diagrams of cloud distributions of selected experiments to see their temporal evolution.

- Have you tried more advection schemes besides: upwind, mpdata 2, mpdata 3?

- You should point out the clear improvement of MPDATA compared to upwind scheme.

- What are the initial conditions in your simulation? What noise do you use?

- Please provide a comparison for the plot of mixing ratios versus $R_d$ (Fig. 9) to observations from nature or experiments. Again, state explicitly in the caption of Fig. 9, what $R_d$ symbolizes: Radius of …

- The error overview plots **Fig. A1ff** are a very interesting way to compare advection schemes and experiments, efficiently. Why do you not pull those into the results section for selected experiments? Is the truth for computing the error the analytical model?

- You should point out in the conclusion, that this study can be a basis for future work.

I am looking forward to providing more detailed comments in the next iteration of this manuscript. For now, some technical comments:

(a) **Fig.1:** ... those are cloud droplet distributions, right? Please state this, explicitly.
(b) **Fig.1:** Can you provide the analytical functions for the distributions & its derivations?
(c) A better description of the Python library is required for interested readers to repeat your experiments. You can do that either in README file on github or in a section of this paper.

---

## Author Comment (AC1)

**GMD manuscript #2020-404: replies to the reviews**

Michael Olesik, Sylwester Arabas and co-authors

**July 2021**

First, let us thank all three reviewers for their valuable critique and feedback. Below, we provide point-by-point replies to all comments. Quoted text of the reviews is typeset in teal and with larger margins.

A revised manuscript with changes automatically highlighted using the latexdiff tool is enclosed with additions marked with blue color and deleted text in red. Additions to the bibliography are not color-highlighted, but all references to new bibliography items are within blue-underlined blocks in the text.

The key change in the manuscript is the introduction of an entirely new section (numbered 3 in the revised manuscript) covering two-dimensional simulations in which one of the dimension corresponds to the particle size and the other to vertical displacement in a column of air. These simulations feature coupling between particle growth and ambient supersaturation dynamics. This addresses the main point of all three reviews of overly simplified test case employed.

**comments by Josef Schröttle, 24 Feb 2021**

This study introduces a new Python library for advection of geophysical flows with the MPDATA scheme. More specifically, it concentrates on the broadening of cloud droplet distributions due to advection and compares those distributions to analytically derived functions. It is based on previous work by the authors in a C++ library and numerous studies that have been conducted since the 1980s up to now. The paper is clearly written in the introduction, the methodology, and results sections. The reader would profit from a more fluent overview of the background literature in this work (in section 1.2),

Several correction to punctuation, the usage of articles and sentence shortening has hopefully improved Section 1.2.

... as well as a brief motivation why this is such important work especially in the context of clouds. A short suggestion of how to incorporate this is given by the reviewer. Also, a brief description of the software for interested users would be very helpful. Overall, this work is unique in its focus on comparing 2-3 advection

schemes in the context of cloud dynamics and could be a basis for many future applications after major revision.

First general comments from my side:

•To give a brief motivation, why your work is so important, I would refer to clouds in climate research, e.g.: Climate goals and computing the future of clouds, Nature Climate Change volume 7, pages 3–5 (2017) or a more recent publication

The background section was extended following the reviewer's suggestion including a reference to the Schneider et al. 2017 paper.

•How do your simulations evolve in time? Besides the cloud distributions you show in Fig. 1ff, I would like to see Hovmoeller diagrams of cloud distributions of selected experiments to see their temporal evolution.

The new section covering two-dimensional simulations features two figures addressing this point. The newly introduced Figure 11 depicts the temporal evolution of the spectral width parameter d, while the new Figure 10 depicts actual binned size spectra at each level of the vertical grid conveying analogous information as a Hovmöller spectral-temporal diagram.

•Have you tried more advection schemes besides: upwind, mpdata 2, mpdata 3?

A comparison with other techniques, in particular with the Lagrangian (moving-sectional, particle-reolved) representation is planned for a follow up study. The concept for this paper is to constitute a guide across different aspects of the MPDATA algorithm which need to (or can optionally) be, taken into account while addressing the particular problem of particle condensational growth.

•You should point out the clear improvement of MPDATA compared to upwind scheme.

The newly introduced Fig. 12 apply highlights this improvement depicting the robustness in which application of even a single corrective iteration of MPDATA basically halves the spectral width of the simulated droplet spectra.

•What are the initial conditions in your simulation? What noise do you use?

The initial condition has been specified in section 1.5, in particular through the equation 1.6 (initial particle spectrum) and the numerical parameters given below (timestep, grid layout). No noise considered.

•Please provide a comparison for the plot of mixing ratios versus Rd (Fig. 9) to observations from nature or experiments. Again, state explicitly in the caption of Fig. 9, what Rd symbolizes: Radius of ...

 $R_d$  symbolizes the analytical-to-numerical ratio of the values of the droplet spectral width d. The misleading  $R_{\text{disp}}$  vs.  $R_d$  naming has been unified and the figure cation includes the definition now instead of a reference to an equation.

The discussion of the newly introduced Figure 11 features a reference to observations.

•The error overview plots Fig. A1ff are a very interesting way to compare advection schemes and experiments, efficiently. Why do you not pull those into the results section for selected experiments? Is the truth for computing the error the analytical model?

Yes, the error measure is based on the analytical solution (as indicated in eq. A1). The highlight of the polar plots presented in the appendix is the Courant number dependence of the rate of convergence. It is of importance when devising case study setups or developing adaptive timestepping criteria - both relevant, yet of secondary relevance to the storyline of the paper, hence presented in an appendix.

•You should - point out in the conclusion, that this study can be a basis for future work.

The last paragraph of the conclusions now outlines the path towards four-dimensional MP-DATA solver capable of integrating bin microphysics dynamics in 3D CFD framework.

I am looking forward to providing more detailed comments in the next iteration of this manuscript. For now, some technical comments:

(a) Fig.1:... those are cloud droplet distributions, right? Please state this, explicitly.

It is now clarified in the caption that the plot depicts particle number densities.

(b) Fig.1: Can you provide the analytical functions for the distributions & its derivations?

It is given with an outline of derivation when introducing eq. 2.5, with reference to the .

(c) A better description of the Python library is required for interested readers to repeat your experiments. You can do that either in README file on github or in a section of this paper.

The PyMPDATA README file has undergone significant expansion including addition of new examples and inclusion of sample code in Julia and Matlab. The public API of the library is now published along with annotations at https://atmos-cloud-sim-uj.github. io/PyMPDATA. Submission of a short paper to the Journal of Open Source Software outlining the package features is planned.

**comments by Anonymous Referee #2, 01 May 2021**

This study examined performance of various MPDATA variants in solving drop size distribution evolution by condensation. The authors reviewed many previous studies in the context of improving MPDATA and showed that MPDATA with three anti-diffusive iterations, third order term, infinite gauge, and the nonoscillatory option reduces the numerical diffusion to roughly a tenth compared to that of the upwind scheme, although it requires  $\sim 10$  times longer than the upwind scheme.

The computational cost footprint is now highlighted in the abstract to clarify that the increased accuracy comes at a trade off.

Although this study examined the performance of MPDATA variants systematically, I would raise two serious problems this study bears. At the current stage, my recommendation is to reject the manuscript for publishing on GMD, and encouraging the authors to improve the manuscript accordingly.

1. Somewhat outdated

I can find several recent papers closely related to the topic this study focuses on: Morrison et al. (2018, doi: 10.1175/JAS-D-18-0055.1), Pardo et al. (2020, doi:10.1175/JAS-D-20-0099.1), and Lee et al. (2021, doi:10.1175/JAS-D-20-0213.1). All those papers already pointed out that drop condensation itself can be sufficiently converged with better schemes or better designed grids, but it is the condensation w/ vertical advection or w/ collision-coalescence that causes serious problems. Furthermore, those studies utilized LES model results in explaining their results, whereas this study only showed the box model results. This study clearly exhibited the performance of MPDATA variants in solving drop condensation, but only the convergence test in solving drop condensation is somewhat outdated compared to the studies I mentioned. I strongly suggest the authors to improve their study by including vertical advection, collision-coalescence, and/or something we do not know its effects.

The newly added section featuring single-column case study involves vertical advection. All three recent papers mentioned are now referenced commented on.

**2. Experimental setting**

In the authors' experimental setting, supersaturation is fixed so the liquid water content increases up to 10 g kg–1, which is almost unrealistic except for tropical cyclones. I strongly suggest the authors to modify the experimental setting so the results become more realistic. For example, Morrison et al. (2018) and Lee et al. (2021) fixed the vertical velocity to be 1 m s–1 for 20 min rather than fixed the supersaturation.

The newly added single-column test case features supersaturation dynamics coupling with the particle growth.

The original box model setup was kept as is. Despite somewhat unrealistic range of liquid water content it allows for performing the convergence analysis across a wide range of gridand timesteps.

**comments by Anonymous Referee #3, 04 May 2021**

This manuscript examines the fidelity of various flavors of the MPDATA advection scheme for solving condensation of the drop size distribution. The writing style is clear and concise, the historical review of bin scheme and MPDATA development was illuminating, and the figures were simple and easy to understand. That said, the study suffers from a few key flaws that lead me to suggest the paper be rejected.

First, the authors recognize that the difficulty of numerically modeling condensation/evaporation is that drop growth processes (in the mass dimension) are fundamentally coupled to spatial advection. Yet the test case, which if I understand correctly was chosen because a reference analytical solution can be obtained, either did not include a spatial advection component or this was not discussed. Morrison et al. (2018) and Lee et al. (2021) both point out that satisfactory solutions can be obtained by a number of schemes in the absence of transport; it is when they are coupled that special consideration must be taken.

The newly added section (number 3 in the revised manuscript) addresses this point by considering a two-dimensional problem with both the spectral and spatial transport considered simultaneously.

Secondly, the physical feasibility of the test case is dubious; in any warm cloud, a mass mixing ratio of 10 g/kg is nigh impossible.

The choice of the test case for the box model simulations was motivated partly by the aim of performing the convergence analysis presented in the appendix which covers a wide range of grid- and timesteps.

Finally, another important aspect (in particular, of Lee et al., 2021) was not covered: the effect of refining grid spacing vs. refining algorithm formulation.

This point is addressed in the analysis presented in the newly introduced Fig. 12 where results with different number of MPDATA iterations are presented for an array of  $\Delta t$ ,  $\Delta r$  and  $\Delta z$  settings.

These three factors combined leave me with the impression that this study, while rigorous, is not relevant to the current state of the field.

I strongly encourage the authors to reconsider the paper by formulating a test case that would demonstrate the relevance of the algorithms tested in dynamical models, and evaluating the trade-off of increased algorithmic accuracy versus refined size grid.

We have followed the request introducing the single-column test case which covers both the vertical transport aspect as well as the supersaturation dynamics coupling.

Let us close this reply by expressing again our thanks for the reviewers' feedback what is also expressed in the acknowledgments section in the revised text.

**On numerical broadening of particle size spectra: a condensational growth study using PyMPDATA 1.0**

Michael Olesik1, Sylwester Arabas1, Jakub Banaśkiewicz1, Piotr Bartman1, Manuel Baumgartner2,3, and Simon Unterstrasser4

1Jagiellonian University, Kraków, Poland

[revised manuscript text omitted]

---

## Referee Report (RR1)

Comments to the manuscript with MS No. gmd-2020-404

General comments:

The authors explored the application of the Multidimensional Positive Definite Advection Transport Algorithm (MPDATA) in simulating the condensational growth of cloud droplets. They have demonstrated that the MPDATA is able to reduce the artificial diffusion of bin-method at the cost of computational resources.

The upwind scheme should be equivalent to a high-order scheme. What is the advantage of using upwind scheme instead of high-order schemes, computational cost?

The authors have demonstrated that the "best variant" is 10 times more costly than the upwind scheme, how feasible is it to use the "best variant" in 3-D cloud models?

This manuscript reads like an early draft that is far from being ready to be peer-reviewed.

To improve the readability of the manuscript, I would suggest the authors summarize the key points of those cited papers and merge those paragraphs that only consists of two sentences in the introduction. Also, the English writing of this manuscript can be improved substantially. I've listed several examples that can be improved in the instruction. But the authors should go through the entire manuscript and try to make the writing more concise.

Theories and numerical methods for "condensational growth of cloud droplets" is well-established, which are even validated against the laboratory experiment. I would suggest the authors consider using common nomenclatures such that the work can be understood by a wide range of audiences in the cloud-physics community.

I **cannot** recommend the publication in its current form. A major revision is required for further consideration.

Specific comments:
1. L.80: Why is the Eulerian scheme is robust in representing particle collisions?
   First, the Eulerian scheme is a mean-field approach, which is not able to represent fluctuations during collisions.
   Second, it is suffering from artificial numerical diffusion in modelling the collision process (Li. et al, 2017).
2. L.90-91: I don't understand the logic of transitioning to Kelvin effect here. Could you elaborate more on this transition?
3. L.112: Do you mean "more significant .... than ..."?
4. L.115-117: I couldn't find the verb of this sentence. Could you please rewrite it?
5. L.125: Please check the grammar.
6. L.132: ... p to x with x being ...

7. L.149: Do you mean "as the following"?
8. L.151: What is "GC" exactly? Is "G" the same as the
9. L.152: Why is the Courant number is "the velocity …"? Do you mean "… to warrant … "?
10. L.161-163: Could you please reword this sentence to improve the readability? For example, we can use short sentences and connect them in a logical way?
11. L.169: You may define what the "S" is first.
12. L.174: Please cite the reference where you got these number.
13. L.186: What is "$(1; 26)$ μm"?
14. L.187: Please check the grammar. Do you mean "in the first …"?
15. L.188: the second. What is "$in\ the\ range\ of\ (0.03; 0.07)$"?
16. L.189: Please rewrite the sentence and make it clearer and avoid mistakes in grammar.
17. L.191: shows.
18. L293: Double-Pass Donor-Cell (DPDC). Please check other parts of the manuscript thoroughly.
19. L.381: If the "best variant" is 10 times more costly than the upwind scheme, how feasible is it to use such a scheme?

Technical corrections:

20. In the title and other parts of the paper, please replace "particle size spectra" by "particle-size spectra" as only two nouns are allowed to be connected at a time in English grammar.
21. MPDATA (Multidimensional Positive Definite Advection Transport Algorithm) should already be fully spelled out in the abstract.
22. Please use consistent key terminologies to improve readability, such as "particle-size distribution" or "particle-size spectra".
23. L.45: Shouldn't the reference be in a bracket?
24. L.47: large-scale models; climate-timescale simulation
25. L.48 : What does "there" refer to? Please check the grammar in this sentence.
26. L.49: What is "particle size-spectrum dynamics "?
27. L.50: What is "size effects"? Do you mean simulation domain-size? Also, please check the grammar of this sentence. For example, "…that…are"?
28. L.53: a population.
29. L.58: What is "the effects"?
30. L.62: "see"→"we refer to". Please also check the format of citations in this sentence.
31. L.63: the application.
32. L.71: size distribution of cloud droplet. Please see my comment 1 in this section.
33. L.79: large-scale.
34. L.81-84: I suggest the authors to shorten the sentences such that they are concise and easy to read.
35. L.86: … can likely be…
36. L.88: Brown (1980) also covers …
37. L.90-91: due to its …

38. L.96: "which is focused on the evaporation of an "aerosol cloud"" does not read right. Also, what is "aerosol cloud"?
39. L.98: compared to.
40. L.99: was used.
41. L.102: Tsang and Rao (1988) pointed out that …the upwind scheme…the prediction accuracy of the mean radius.
42. L.105: …in a chapter focusing…
43. L.106: was presented.
44. L.108: What is "particle size computations"? The latter lists …
45. L.110: …the condensational growth…
46. L.120: what is "sic!"?

**References**

Li, X.-Y., Brandenburg, A., Haugen, N. E. L., and Svensson, G.: Eulerian and Lagrangian approaches to multidimensional condensation and collection, J. Adv. Model. Earth Syst., 9, https://doi.org/10.1002/2017MS000930, 2017.

---

## Referee Report (RR2)

Comments to the manuscript with MS No. gmd-2020-404

General comments:

1. The authors have improved the presentation of the manuscript substantially in this revision. However, it is still hard to read. It took me more time to understand and comment on the writing instead of science.
I suggest all the authors read through the manuscript carefully to improve the readability.
I would also encourage the authors to explain each figure clearly and logically.
Please see my suggestions in the section of specific comments.

2. It is still unclear to me what the advantage of using the upwind scheme over high-order schemes is. How is the computational cost of this scheme compared to the high-order schemes? Discussing this briefly in the introduction and the discussion part can be beneficial for the community.
Indeed, in the result section, the authors argue that "… As a result, the algorithm is characterised by reduced numerical diffusion while maintaining the salient features of the underly …". Which benchmark scheme did you compare with?

3. The authors found that "… even a tenfold decrease of the spurious numerical spectral broadening can be obtained by an apt choice of the MPDATA variant (maintaining the same spatial and temporal resolution), yet at an increased computational cost …". In my opinion, this is an interesting finding since it helps understanding the artificial broadening of particle-size distribution. What is the benchmark for this conclusion?

I am not asking for performing more simulations. Comparing your current results to some references should be enough to address the question.

4. Is the "box model simulation" direct numerical simulation? What is your numerical setup? Is it a 1-D case driven by a constant supersaturation? What are the governing equations of the motion of particles? How do you determine your time step? What are the boundary conditions?

5. What is the link between the "box model simulation" and the single-column model?
Shouldn't the results from "box model simulation" be used or compared to the sub-grid scale modeling of large-eddy simulations before jumping to the single-column model?
Can the conclusion of MPDATA from the box simulation be carried over to the single-column simulation? I encourage the authors to discuss this link in depth, which can be a highlight of this study.

Again, I **cannot** recommend the publication in its current form. A major revision is required for further consideration.

Specific comments:

1. P1, L5: Could you please rewrite "The numerical diffusion problem inherent to the employment of the fixed-bin discretisation in the numerical solution of the arising transport problem is scrutinised." to make it more concise?

2. P1, L5: What is "carried out"? Neither "Eulerian modelling approach" nor "evolution of the probability …" fit the subject.

3. P1, L25: What is a "The single-column problem"?

4. P1, L51: What is "a population-balance equation"? Please add references. Isn't it the Boltzmann transportation equation conserving mass?

5. P2, L10: Do you mean "turbulent mixing"? Numerical approach?

6. P2, L15: Please be more specific about the challenging physical processes. What is "represent the subtleties"?

7. P2, L20: Please cite "https://doi.org/10.1146/annurev-fluid-011212-140750". "inherent limitations" of what?

8. P2, L35: in determining both the …

9. P2, L40: Please add references to the statement "The parameterisations used in climate models are developed based on smaller-scale simulations resolving particle-size spectrum evolution.". Small-scale models can simulate the cloud microphysical processes while GCMs cannot. Can you give an example on which GCM adopts processes simulated using which small-scale model?

10. P2, L75: What is "title ="? What do you mean here?

11. P2, L105: droplets?

12. P3, L5: by comparing to …

13. P3, L5: the upwind …

14. P3, L25: praised. What do you mean by "prediction accuracy of …"?

15. How do I connect the paragraphs above and below the paragraph "Aerosol Science: …"? What is the scientific question you want to summarize from this book?

16. P3, L40: I don't understand this sentence. Please rephrase.

17. P3, L45: fixed and moving bin-approaches

18. P3, L50: What do you mean by "…at.., …at…"?

19: What do you mean by "a grid composed of 2000 size bins"? Is it the spatial resolution?

20: P3, L90: I don't understand the statement of "there is a degree of freedom in the choice of the particle-size parameter used as the coordinate". $n(r)$, $n(s)$, and $n(v)$ are exchangeable. It is a matter of preference of using the radius binning or mass binning. How is it related to "a degree of freedom"? The same comment applies to "$p(r(x))$".

21: P4, L20: The sentence is incomplete.

22: Caption of Fig.1: What is "$\ln 2(r^3)$"? To improve the readability of Fig.1 and 2, you could move the title to the y-axis and use the abbreviation "m, cm, um" instead of words. Please check all the figures and improve the labels and units presented in them.
I still don't understand the purpose of showing both Fig.1 and 2. Fig.2 looks almost identical to Fig.1. If you want to compare the two cases, you should plot them in a same figure.

23: P5, L10: Which figures are you talking about? This paragraph is repeating what is said in the caption of Fig.1. What is the point of repeating it?

24: P5, L15: What do you mean by saying "integrating the number conservation law"?

25: At least the last three paragraphs of section 2.2 can be merged into one. You may describe and explain the results in a logical way. This is just one example, please carefully read through the description of other figures and make the storyline fluent and concise.

26: P6, L5: Which "resulting in" which?

27: P6, L10: What is "sought modified equation"?

28: P6, L55: What do you mean by "transported signal" and "variable sign signals."? Numerical formula. Please rephrase this sentence. The grammar is not correct.

29: P6, L65: About this statement "Overall, while the MPDATA solutions are superior to upwind, the drop in amplitude and broadening of the resultant spectrum still visibly differs from the discretised analytical solution.", is it because of the hyper-diffusion method used in MPDATA or the bin resolution? The upwind scheme presented in Fig.1 of "**https://doi.org/10.1002/2017MS000930**" also underrepresents the amplitude of peaks.

30: P7, L5: What do you mean by "linearising MPDATA about an arbitrarily large constant"? It should be "background scalar field".

31: P7, L15: "…, such". It should be a new sentence "Such …".

32: You can combine Fig 1, 2, 3, and 4 into one figure to improve the readability. Please go through the manuscript carefully and make the figures concise.

33: P7, L25: Are the "negative values" generic of the "infinite gauge" or just for your case? Please explain why they are negative.

34: The last 3 paragraphs in the section 2.11 can be merged into one.

35: P10, L10: "right panel"→ "rhs panel". Please check "right" and "left" expressions all across the manuscript.

36: P10, L20: "so"→"such that"

37: P10, L25, L30: I don't understand this paragraph. Please check the grammar.

38: P11, L10: "what"→"that"

39: P11, L55: What is the "Ordinary particle volume concentration"?

40: P12, L30: Is "w1" the initial vertical velocity?

41: P12, L75: How is the vertical velocity determined?

42: The caption of Fig.10 does not read right.

43: The first paragraph of section 3.3 describes the numerical setup. It can be moved up to the one-sentence paragraph.

44: Fig.11: It is hard to distinguish different lines in the vertical-profile plots. Can you use different symbols? Why does "MPDATA iterations: 1" yield different d compared with "MPDATA iterations: 2" and "MPDATA iterations: 3".

45: P13, L40: "the only conclusion here is that the visualisation method used in Fig. 10 is apt to highlight this feature". This statement again questions the application of the single-column simulation in this manuscript. I still don't understand if the conclusion about the MPDATA from the box simulation can be carried over to the single-column simulation. I encourage the authors to discuss this link in depth, which can be a highlight of this study. Otherwise, I don't see the point of including a very simplified simulation of the single-column model.

46. P15, L15: This study focuses on …

---

## Referee Report (RR3)

Comments to the manuscript with MS No. gmd-2020-404

General comments:

We thank the authors for improving the presentation of the manuscript and for satisfactorily addressing the comments. I recommend the publication of the manuscript after very minor revision.

Specific comments:
    1. L25: physical-chemical

---

## Author Response (AR2)

**GMD manuscript #2020-404:**
**replies to the second round of reviews**

Michael Olesik, Sylwester Arabas and co-authors

October 2021

We hereby express our appreciation for the reviewers' and editor's feedback.
Below, we include point-by-point replies to all comments provided by the reviewers. Quoted text of the reviews is typeset in teal and with larger margins.
A revised manuscript with changes automatically highlighted using the latexdiff tool is enclosed with additions marked with blue colour and deleted text in red. Additions to the bibliography are not colour-highlighted, but all references to new bibliography items are within blue-underlined blocks in the text.

**comments by Anonymous Referee #2, 31 Aug 2021**

> Second review for the manuscript [gmd-2020-404] entitled "On numerical broadening of particle size spectra: a condensational growth study using PyMPDATA 1.0" written by Olesik et al.

> On the first stage, as a reviewer, I requested the authors to improve the manuscript by performing some additional experiments, because some studies have already shown the importance of vertical advection in the broadening of drop size spectra during condensational growth. The authors did perform extra experiments in which the vertical advection of drops and water vapor mixing ratio are allowed, and added a section to the manuscript. In their experiments, vertical grid spacing is the most important parameter in controlling the DSD broadening, as already pointed out by Morrison et al. (2018) and Lee et al. (2021). Although they added the results including the effects of vertical advection, however, this study still contains some serious problems described below. My recommendation is to reject this manuscript for publishing on GMD.

> First, there are few links between the box model study (Section 2) and the column model study (Section 3). The two studies use totally different environmental setups, and furthermore, while the box model study focuses on the variant MPDATA flavors, the column model study fixes the numerical method (except the

*number of iterations) and just focuses on the sensitivity on experimental parameters such as time step, vertical grid spacing, and spectral spacing. I do not feel that this manuscript is well organized, and therefore, it is hard to understand what the authors want to say.*

The two main goals of the paper can be summarised as follows. First, to comment on several studies where unspecified variants of MPDATA were applied for calculations of condensational growth and led to misleading general statements on the performance of the scheme. Second, but also to support the first point, we aim at providing an abridged tour of MPDATA features and nuances relevant to the problem of condensational growth. To this end, the box-model part (Section 2) features a simplest possible test case enabling depiction of the algorithm options and characteristics. All figures presented in the paper are readily reproducible using pure-Python open-source code aiding potential developers in building upon the presented developments. Revising the paper, we aimed at improving the flow, yet the concept of two separate sections remained.

*Second, the relative dispersions presented in this study have a parabolic shape, minimum in the middle of column. I cannot find such a shape in any other previous studies, even in the reference the authors mentioned (Arabas et al. 2009). I am not sure whether the parabolic relative dispersion is attributable for their imperfect experimental settings such as excluding the vertical advection of potential temperature, but it is clear that the authors should be more careful in explaining the results.*

As an example, in figures 2, 3 & 6 in Lu and Seinfeld (2006)[1], which are based on LES simulations of with bin microphysics, the "parabolic" shape is also evident. The hypotheses put forward by Lu and Seinfeld (2006) are that:

- *"the peaks of d at cloud top are most likely due to the entrainment mixing of quiescent unsaturated free tropospheric air with saturated cloudy air"*;

- *"the peaks of d and $\sigma$ near the cloud base are a result of "entity" mixing, as described by Telford et al. [1984]."*;

- *"the minima of d and $\sigma$ at midcloud are the result of competition between downdraft broadening and condensational growth narrowing."*.

The employed single-column kinematic framework clearly does not include the processes featured in the above argumentation, what in fact is interesting from the point of view of validating these hypotheses. Yet, as it has been clearly indicated in the manuscript, the main point in comparing the obtained profiles with results from more complex simulations and measurements is to confirm if the test case covers the parameter space relevant to the studied problem. A reference to the discussion in Lu and Seinfeld (2006) has been added.
* * *
[1] https://doi.org/10.1029/2005JD006419

Third, I feel that the manuscript includes too many rhetoric and unnecessary conjunctions, which make the manuscript difficult to read.

We have aimed at correcting it by shortening sentences and being more specific in numerous statements.

As for minor problems, 1) Figure 10 is very hard to understand, and 2) the authors seem to have made some mistakes in controlling the experimental parameters to make Figure 12 (e.g., $\Delta t$ in the second column of the first row and the third column of the third row).

Figure 10 is a three-dimensional perspective view of a snapshot of the simulation state. It is only meant to provide a qualitative and hopefully intuitive picture of the modelled system. The discussion of the simulation results is based on subsequent two-dimensional plots in figures 11 and 12.
It has been indicated in the text that the timestep for each run is adjusted to maintain stability constraint of the scheme. The value indicated in each plot is the actual value used after the adjustment.

**comments by Anonymous Referee #4, 3 Sep 2021**

General comments:

The authors explored the application of the Multidimensional Positive Definite Advection Transport Algorithm (MPDATA) in simulating the condensational growth of cloud droplets. They have demonstrated that the MPDATA is able to reduce the artificial diffusion of bin-method at the cost of computational resources.

The upwind scheme should be equivalent to a high-order scheme. What is the advantage of using upwind scheme instead of high-order schemes, computational cost?

MPDATA relies on iterative application of upwind with the first iteration employing physical velocity and subsequent corrective iterations using antidiffusive velocities. The resultant scheme, depending on the variant chosen, is second- or third-order in time and space, while maintaining the key properties of upwind, namely positive-definiteness and stability constraints.

The authors have demonstrated that the "best variant" is 10 times more costly than the upwind scheme, how feasible is it to use the "best variant" in 3-D cloud models?

In the section on computational cost in the revised manuscript, we added references to the studies of Liu et al. (1997)[2] and Onishi et al. (2010)[3] where analogous upwind-normalised measures were reported. The methods analysed in these two studies had roughly three and four times higher cost than upwind, and these methods are used in 3-D cloud models. Clearly, the computational cost is not the only relevant measure: memory footprint, parallelisation opportunities, convergence rate, timestep constraints, monotonicity – all these factors need to be taken into account for a comprehensive assessment of a scheme, what is now also underlined in the discussion of the computational cost.

> This manuscript reads like an early draft that is far from being ready to be peer-reviewed. To improve the readability of the manuscript, I would suggest the authors summarize the key points of those cited papers and merge those paragraphs that only consists of two sentences in the introduction. Also, the English writing of this manuscript can be improved substantially. I've listed several examples that can be improved in the instruction. But the authors should go through the entire manuscript and try to make the writing more concise.

We have addressed the points raised by both reviewers in this regard and redacted multiple passages aiming at correcting the language and improving conciseness.

> Theories and numerical methods for "condensational growth of cloud droplets" is well-established, which are even validated against the laboratory experiment. I would suggest the authors consider using common nomenclatures such that the work can be understood by a wide range of audiences in the cloud-physics community.
>
> I cannot recommend the publication in its current form. A major revision is required for further consideration.

The manuscript has undergone a major revision. While we do aim at catering to a wide audience range, our key goal remains to provide a detailed set of instructions for geoscientific model developers - in line with the journal scope. To this end, we propose to retain within the manuscript discussion of such aspects as handing mass-doubling coordinate transformation within the upwind numerics, an aspect that on the one hand seems basic, on the other hand has not been explicitly covered in earlier works on the topic. Having picked test cases from literature, we hope that further links with the common nomenclature can also be obtained through the referenced works.

> Specific comments: 1. L.80: Why is the Eulerian scheme is robust in representing particle collisions? First, the Eulerian scheme is a mean-field approach, which is
* * *
[2]https://doi.org/10.1175/1520-0469(1997)054<2493:VOMFCO>2.0.CO;2
[3]https://doi.org/10.1299/jee.5.1

> not able to represent fluctuations during collisions. Second, it is suffering from artificial numerical diffusion in modelling the collision process (Li. et al, 2017)[4].

This sentence was removed. The two aspects the Reviewer point out are certainly true. There are also aspects which make bin representation "robust" such as consistency of response to subgrid dynamics in a CFD solver or simply implementation aspects, yet we opt not to discuss it in the present paper (the reference to Li et al. 2017 has been cited in the Background section).

> 2. L.90-91: I don't understand the logic of transitioning to Kelvin effect here. Could you elaborate more on this transition?

Added to the discussion of eq. 2.3.

> 3. L.112: Do you mean "more significant .... than ..."?

Corrected.

> 4. L.115-117: I couldn't find the verb of this sentence. Could you please rewrite it?

Rephrased.

> 5. L.125: Please check the grammar.

Corrected and shortened.

> 6. L.132: ... p to x with x being ...

Changed.

> 7. L.149: Do you mean "as the following"?

Rephrased.

> 8. L.151: What is "GC" exactly? Is "G" the same as the

This is now clarified in a newly added sentence: "*Note that the values of the Courant number itself are not used, only the product GC of the coordinate transformation term G and the Courant number C.*"
* * *
[4]Li, X.-Y., Brandenburg, A., Haugen, N. E. L., and Svensson, G.: Eulerian and Lagrangian approaches to multidimensional condensation and collection, J. Adv. Model. Earth Syst., 9, `https://doi.org/10.1002/2017MS000930`, 2017.

9. L.152: Why is the Courant number is "the velocity . . . "? Do you mean ". . . to warrant . . . "?

Simplified the sentence.

10. L.161-163: Could you please reword this sentence to improve the readability? For example, we can use short sentences and connect them in a logical way?

Shortened.

11. L.169: You may define what the "S" is first.

Added clarification referring to relatie humidity.

12. L.174: Please cite the reference where you got these number.

Added reference to East & Marshall 1954[5].

13. L.186: What is "(1; 26) $\mu$m"?

Shortened and rephrased: "*The domain span is 1–26 $\mu$ m.*".

14. L.187: Please check the grammar. Do you mean "in the first . . . "?

Rephrased.

15. L.188: the second. What is "in the range of (0.03; 0.07)"?

Rephrased: "*variable Courant number approximately in the range of 0.03 to 0.07 for $p = r$*"

16. L.189: Please rewrite the sentence and make it clearer and avoid mistakes in grammar.

Rewritten.

17. L.191: shows.

Corrected (made the subject plural).

18. L293: Double-Pass Donor-Cell (DPDC). Please check other parts of the manuscript thoroughly.

Spelled as suggested, also reserved the term "pass" for DPDC now "iterations" are used whenever generally referring to MPDATA iterations.
* * *
[5]https://doi.org/10.1002/qj.49708034305

19. L.381: If the "best variant" is 10 times more costly than the upwind scheme, how feasible is it to use such a scheme?

The use of upwind as a benchmark...

Technical corrections:

20. In the title and other parts of the paper, please replace "particle size spectra" by "particle-size spectra" as only two nouns are allowed to be connected at a time in English grammar.

Corrected.

21. MPDATA (Multidimensional Positive Definite Advection Transport Algorithm) should already be fully spelled out in the abstract.

Corrected.

22. Please use consistent key terminologies to improve readability, such as "particle-size distribution" or "particle-size spectra".

Corrected.

Corrected.

24. L.47: large-scale models; climate-timescale simulation

Both corrected.

25. L.48 : What does "there" refer to? Please check the grammar in this sentence.

"there" removed.

Rephrased with "*resolving particle-size spectrum evolution*".

27. L.50: What is "size effects"? Do you mean simulation domain-size? Also, please check the grammar of this sentence. For example, "...that...are"?

Surplus "size" removed, sentence rephrased.

Corrected.

29. L.58: What is "the effects"?

Rephrased removing the word "effect".

30. L.62: "see" − > "we refer to". Please also check the format of citations in this sentence.

Changed and fixed the citation enumeration.

31. L.63: the application.

Corrected.

32. L.71: size distribution of cloud droplet. Please see my comment 1 in this section.

Changed to "droplet-size spectrum".

33. L.79: large-scale.

Corrected.

34. L.81-84: I suggest the authors to shorten the sentences such that they are concise and easy to read.

Shortened and split in two.

35. L.86: . . . can likely be. . .

Changed.

36. L.88: Brown (1980) also covers . . .

Changed.

37. L.90-91: due to its . . .

Changed.

38. L.96: "which is focused on the evaporation of an "aerosol cloud"" does not read right. Also, what is "aerosol cloud"?

Shortened to "which is focused on evaporation".

39. L.98: compared to.

Corrected (in several other places as well).

40. L.99: was used.

Corrected.

41. L.102: Tsang and Rao (1988) pointed out that . . . the upwind scheme. . . the prediction accuracy of the mean radius.

Changed as suggested.

42. L.105: . . . in a chapter focusing. . .

Corrected.

43. L.106: was presented.

Corrected.

44. L.108: What is "particle size computations"? The latter lists . . .

Rephrased (and corrected latter to first as it was swapped).

45. L.110: . . . the condensational growth. . .

Corrected.

46. L.120: what is "sic!"?

Removed for clarity (was meant to underline no typo despite large value reported)

---

## Author Response (AR3)

**GMD manuscript #2020-404:**
**reply in the third round of review**

**Michael Olesik, Sylwester Arabas and co-authors**

**January 2022**

We hereby express our appreciation for the reviewer's feedback which we followd point-by-point revising the text. Below, we include replies to all points. Quoted text of the review is typeset in teal and with larger margins.

A revised manuscript with changes automatically highlighted using the latexdiff tool is enclosed with additions marked with blue colour and deleted text in red. Additions to the bibliography are not colour-highlighted, but all references to new bibliography items are within blue-underlined blocks in the text.

**comments by Anonymous Referee #4, 01 Dec 2021**

> General comments:
> 1. The authors have improved the presentation of the manuscript substantially in this revision. However, it is still hard to read. It took me more time to understand and comment on the writing instead of science. I suggest all the authors read through the manuscript carefully to improve the readability. I would also encourage the authors to explain each figure clearly and logically. Please see my suggestions in the section of specific comments.

We have revised the manuscript following the provided suggestions. Additionally, numerous language corrections aimed at improving flow in the text were introduced.

> 2. It is still unclear to me what the advantage of using the upwind scheme over high-order schemes is. How is the computational cost of this scheme compared to the high-order schemes? Discussing this briefly in the introduction and the discussion part can be beneficial for the community.

MPDATA is a higher-order scheme based on upwind. The advantage of basing the scheme on upwind is that MPDATA retains the salient features of upwind: conservativeness, small phase error, sign preservation. This is now mentioned in the very first paragraph of the paper.

Indeed, in the result section, the authors argue that "...As a result, the algorithm is characterised by reduced numerical diffusion while maintaining the salient features of the underly...". Which benchmark scheme did you compare with?

Upwind, clarified.

3. The authors found that "...even a tenfold decrease of the spurious numerical spectral broadening can be obtained by an apt choice of the MPDATA variant (maintaining the same spatial and temporal resolution), yet at an increased computational cost...". In my opinion, this is an interesting finding since it helps understanding the artificial broadening of particle-size distribution. What is the benchmark for this conclusion?

The benchmark behind the "tenfold" figure is the upwind solution. This has been clearly indicated in the discussion of Figure 9 based on which the statement is made. The passage in the abstract which the Referee refers to is now clarified, and the sentence reads: "... *compared with upwind solutions, even a tenfold decrease ...*".

I am not asking for performing more simulations. Comparing your current results to some references should be enough to address the question.

Throughout the paper, the results from numerical simulations with MPDATA are compared to both analytical solutions (box test case), as well to upwind numerical solutions (box and single-column test cases).

4. Is the "box model simulation" direct numerical simulation? What is your numerical setup? Is it a 1-D case driven by a constant supersaturation? What are the governing equations of the motion of particles? How do you determine your timestep? What are the boundary conditions?

The "box-model" notion was used here as in the chemical/microphysical jargon meaning a model with no spatial dimension, without any fluid flow. Particle positions or motion are not considered at all. The only "flow" is in the size-spectral dimension. The boundary conditions are specified in the text: *linear extrapolation is applied for G, while both $\psi$ and GC are set to zero within the halo*. For the sample simulations discussed in the main text, the timestep is set to $\Delta t = \frac{1}{3}$ s as indicated in the text, while a range of timestep settings is used to perform the convergence analysis presented in the appendix.
To avoid confusion with DNS, the lack of spatial dimension is now verbosely stated in the text: both in the abstract and near the first mention of a box-model problem.

5. What is the link between the "box model simulation" and the single-column model? Shouldn't the results from "box model simulation" be used or compared to the sub-grid scale modeling of large-eddy simulations before jumping to the single-column model? Can the conclusion of MPDATA from the box simulation

be carried over to the single-column simulation? I encourage the authors to discuss this link in depth, which can be a highlight of this study. Again, I cannot recommend the publication in its current form. A major revision is required for further consideration.

Assuming the above comment is based on the assumption of the box model being a kind of direct numerical simulation, let us reiterate that there is no physical motion of any kind considered in the box test case. The single-column setup adds the spatial dimension to the problem. All conclusions concerning spectral advection and spectral broadening carry over to the spatio-spectral problem of the single-column simulations.

Specific comments:
1. P1, L5: Could you please rewrite "The numerical diffusion problem inherent to the employment of the fixed-bin discretisation in the numerical solution of the arising transport problem is scrutinised." to make it more concise?

The first sentences of the abstract were rewritten.

2. P1, L5: What is "carried out"? Neither "Eulerian modelling approach" nor "evolution of the probability ..." fit the subject.

Changed to: "*a fixed-bin discretisation (so-called "bin" microphysics) is used in solution ....*

3. P1, L25: What is a "The single-column problem"?

Rephrased to "*single-column test case*".

4. P1, L51: What is "a population-balance equation"? Please add references. Isn't it the Boltzmann transportation equation conserving mass?

Given that the present work deals only with the condensational growth of a population of particles, a reference to Boltzmann transport equation seems to add little to the presented discussion. While not recalling the connection with Boltzmann formalism to be prominently featured in any of the cited references, let us note that the term "population-balance equation" had been featured even in the title of Tsang & Rao (1990) for describing the very same problem and even the very same numerical solution methods. We thus intend to leave it as is. A reference to a textbook on population balance equations (Ramkrishna 2000) is now added right after presenting eq. 1.1.

5. P2, L10: Do you mean "turbulent mixing"? Numerical approach?

This sentence was meant to juxtapose mixing (physical) effects with numerical solution artefacts. It is now split in two and augmented with more examples of physical processes (turbulent mixing, diverse particle composition, radiative heat transfer effects), with reference to discussion in Feingold and Chuang (2002).

6. P2, L15: Please be more specific about the challenging physical processes. What is "represent the subtleties"?

Added the following: "*which link the physio-chemical properties of single particles with ambient thermodynamics through latent heat release and multi-particle competition for available vapour*".

7. P2, L20: Please cite "https://doi.org/10.1146/annurev-fluid-011212-140750".

This work had already been cited two sentences ahead. All reference parentheses in this introductory paragraph have the "e.g." disclaimer.

"inherent limitations" of what?

Inherent limitations of "discretisation".

8. P2, L35: in determining both the ...

Corrected.

9. P2, L40: Please add references to the statement "The parameterisations used in climate models are developed based on smaller-scale simulations resolving particle-size spectrum evolution.". Small-scale models can simulate the cloud microphysical processes while GCMs cannot. Can you give an example on which GCM adopts processes simulated using which small-scale model?

Representation of cloud condensation nuclei activation in GCMs can be a good example here. Simple parameterisations for use in GCMs are built using detailed parcel-model simulations involving representation of droplet size-spectrum dynamics. A mention of it is now added to the sentence.

10. P2, L75: What is "title ="? What do you mean here?

This was an accidentally included part of a bibliography record – removed.

11. P2, L105: droplets?

Changed "numerical broadening of the spectrum" into "numerical broadening of the cloud droplet spectrum"

12. P3, L5: by comparing to ...

A sentence explaining that the study compared upwind to an Eulerian-Lagrangian schemes is now added.

13. P3, L5:the upwind ...

Added "the".

14.P3, L25: praised. What do you mean by "prediction accuracy of..."?

The word "accuracy" is now removed (as it was indeed not used in the cited work).

15. How do I connect the paragraphs above and below the paragraph "Aerosol Science: ..."? What is the scientific question you want to summarize from this book?

The literature review is presented in chronological order, which is why the reference to Williams and Loyalka 1991 book is given at this point – after Tsang and Rao 1988, and before Kostoglou and Karabelas 1995). The book lists MPDATA among methods (of choice) for solving the condensation term in aerosol population balance equations, which is the reason for citing it here.

16. P3, L40: I don't understand this sentence. Please rephrase.

Rephrased and shortened.

Rephrased.

The entire passage is now rewritten.

19: What do you mean by "a grid composed of 2000 size bins"? Is it the spatial resolution?

Spectral resolution. Rephrased to: "*spectral discretisation involving 2000 size bins*".

20: P3, L90: I don't understand the statement of "there is a degree of freedom in the choice of the particle-size parameter used as the coordinate". n(r), n(s), and n(v) are exchangeable. It is a matter of preference of using the radius binning or mass binning. How is it related to "a degree of freedom"? The same comment applies to "p(r(x))".

By "degree of freedom" we meant "it is a matter of preference", yet since the choice influences the numerical solution characteristics, we would also refrain from calling it just "preference". The two passages are now rephrased and start with "one has the choice" instead of "there is a degree of freedom".

Corrected.

22: Caption of Fig.1: What is "$ln2(r^3)$"?

This is the definition of the mass-doubling grid layout, what is now verbosely stated in the figure caption and the defining equation is given in parenthesis.

To improve the readability of Fig.1 and 2, you could move the title to the y-axis and use the abbreviation "m, cm, um" instead of words. Please check all the figures and improve the labels and units presented in them.

Figures 1 through 8 were re-plotted with abbreviated unit labels.

I still don't understand the purpose of showing both Fig.1 and 2. Fig.2 looks almost identical to Fig.1. If you want to compare the two cases, you should plot them in a same figure.

Both figures include spectra at t=0. These are discretised on different grids – this is the difference between the two figures and the purpose of having both presented in the paper. Merging the two figures would cause the two t=0 histograms to overlap and be illegible.

23: P5, L10: Which figures are you talking about? This paragraph is repeating what is said in the caption of Fig.1. What is the point of repeating it?

Removed.

24: P5, L15: What do you mean by saying "integrating the number conservation law"?

Rephrased to "*solving number conservation equation*".

25: At least the last three paragraphs of section 2.2 can be merged into one. You may describe and explain the results in a logical way. This is just one example, please carefully read through the description of other figures and make the storyline fluent and concise.

Merged and augmented with clarification of linear vs. mass-doubling grid comparison.

26: P6, L5: Which "resulting in" which?

Rephrased to "*what results in*", "*which can be further ...*"

27: P6, L10: What is "sought modified equation"?

In the modified equation analysis outlined in the paragraph in question, one seeks the partial differential equation that is actually represented by a numerical scheme (instead of the original physical equation). In this case, the original physical equation is the advection equation, while the leading terms of the numerically realised modified equation constitute an advection-diffusion equation. The additional diffusion term originates from numerical approximation, hence the term "numerical diffusion".

To clarify the statement, the following sentence is now added at the beginning of the paragraph: "*In a nutshell, the analysis involves: (i) Taylor-expansion of each term of the numerical scheme, (ii) elimination of higher-than-first order time derivatives using the time-differentiated original advection equation, and (iii) derivation of a partial differential equation, referred to as the modified equation, that a given scheme actually approximates in lieu of the advection equation.*"

> 28: P6, L55: What do you mean by "transported signal" and "variable sign signals."? Numerical formula. Please rephrase this sentence. The grammar is not correct.

Split into two sentences, replace "transported signal" with $\psi$, removed "numerical". Also, the term "signal" was replaced with "field" elsewhere.

> 29: P6, L65: About this statement "Overall, while the MPDATA solutions are superior to upwind, the drop in amplitude and broadening of the resultant spectrum still visibly differs from the discretised analytical solution.", is it because of the hyper-diffusion method used in MPDATA or the bin resolution?

Nowhere in the present work, we have used the term "hyper-diffusion". The corrective iterations of MPDATA employ anti-diffusive velocities. The corrections reduce the drop in amplitude. Of course, the bin resolution influences the results, yet in the passage in question, we compare MPDATA and upwind solutions obtained for the same bin resolution.

> The upwind scheme presented in Fig.1 of "https://doi.org/10.1002/2017MS000930" also underrepresents the amplitude of peaks.

Reference to Fig 1a in Li et al. 1997 is now mentioned as well.

> 30: P7, L5: What do you mean by "linearising MPDATA about an arbitrarily large constant"? It should be "background scalar field".

The employed wording is taken from Smolarkiewicz 2006 (also used in Hill 2010). The requested "background scalar field" had already been used in the parenthesis in the very same sentence.

> 31: P7, L15: "..., such". It should be a new sentence "Such ...".

Rephrased and split in two sentences as suggested.

*32: You can combine Fig 1, 2, 3, and 4 into one figure to improve the readability. Please go through the manuscript carefully and make the figures concise.*

Figures 1–8 may certainly appear repetitive, yet this stems from the tutorial character of section 2. Moreover, the discussed features of numerical solution such as: differences in employed grids, appearance of negative values or differences in bin amplitude for different number of MPDATA iterations would be obscured if plotted all on a single figure.

*33: P7, L25: Are the "negative values" generic of the "infinite gauge" or just for your case? Please explain why they are negative.*

The previous paragraph ended with "*such gauge choice decreases the amplitude of the truncation error, however, it makes the algorithm no longer positive definite.*" The above sentence was split in two leaving the statement on lost positive-definite property clearer.

*34: The last 3 paragraphs in the section 2.11 can be merged into one.*

Merged.

*35: P10, L10: "right panel" → "rhs panel". Please check "right" and "left" expressions all across the manuscript.*

In all references to figure panels, "righ-hand panel" and "left-hand panel" are now used.

*36: P10, L20: "so" → "such that"*

Corrected.

*37: P10, L25, L30: I don't understand this paragraph. Please check the grammar.*

The first long sentence was split in two. A reference to more comprehensive discussion of the issue was added (Smolarkiewicz and Rasch 1991).

*38: P11, L10: "what" → "that"*

Corrected.

*39: P11, L55: What is the "Ordinary particle volume concentration"?*

It was meant to underline difference of per-volume concentration vs. specific number concentration (per mass of dry air). The word "ordinary" is now removed and the sentence starts with: "Particle volume concentration (as opposed to specific number concentration)...".

*40: P12, L30: Is "w1 "the initial vertical velocity?*

No, as had been stated in the very same sentence, $w_1$ is a parameter of the equation determining the time evolution of the vertical momentum density: $\rho_d w(z,t) = \rho_d w_1 \sin(\pi t/t_1)(1 - H(t - t_1))$.

> 41: P12, L75: How is the vertical velocity determined?

To clarify, the equation defining the vertical velocity is now numbered (3.1) and referenced.

> 42: The caption of Fig.10 does not read right.

Rephrased.

> 43: The first paragraph of section 3.3 describes the numerical setup. It can be moved up to the one-sentence paragraph.

Moved.

> 44: Fig.11: It is hard to distinguish different lines in the vertical-profile plots. Can you use different symbols?

The different lines are now plotted with different colours in addition to different line styles.

> Why does "MPDATA iterations: 1" yield different d compared with "MPDATA iterations: 2" and "MPDATA iterations: 3".

This is the key message of the paper. The relative dispersion $d$ is the measure of spectrum broadness. MPDATA corrective iterations reduce numerical broadening yielding smaller $d$.

> 45: P13, L40: "the only conclusion here is that the visualisation method used in Fig. 10 is apt to highlight this feature". This statement again questions the application of the single-column simulation in this manuscript.

This statement was indeed unfortunate, and is now replaced by one ending simply with ... *no physical interpretation is warranted*. The single-column KiD framework is certainly very simple, yet it has proven in numerous cited works to be a useful model for works focusing on the basics of cloud microphysics/macrophysics interplay - as it is used herein.

> I still don't understand if the conclusion about the MPDATA from the box simulation can be carried over to the single-column simulation. I encourage the authors to discuss this link in depth, which can be a highlight of this study. Otherwise, I don't see the point of including a very simplified simulation of the single-column model.

This comment is likely based on the incorrect assumption that the box model simulations were a kind of direct-numerical simulation. The box model simulations does not involve any notion of fluid flow or motion of any kind, the only considered transport is that in size-spectral space. The single-column simulations add the spatial dimension, what was explicitly requested in earlier reviews. All conclusions from the box model runs (with just one spectral dimension) carry over to multi-dimensional cases including the single-column case of spatio-spectral (2D) transport.

46. P15, L15: This study focuses on...

Changed.

Thank you.

---

## Author Response (AR4)

**GMD manuscript #2020-404**

**March 2022**

We express our appreciation for the reviewer's feedback and positive overall assessment.
A revised manuscript with changes automatically highlighted using the latexdiff tool is enclosed with additions marked with blue colour and deleted text in red.
We have shifted from providing the code in the electronic supplement to zenodo archives.
We have adjusted the sentence in the abstract pointed out by the reviewer.
We have also added one funding acknowledgement.

Thank you.

Michael Olesik, Sylwester Arabas and co-authors.